

# INSPIRE Game: Integration of vulnerability in impact-based forecasting of urban floods

Akshay Singhal [1,2], Louise Crochemore [2], Isabelle Ruin [2], and Sanjeev K. Jha [1]

[1]Indian Institute of Science Education and Research Bhopal, Madhya Pradesh, India
[2]Univ. Grenoble Alpes, CNRS, INRAE, IRD, Grenoble INP, IGE, 38000 Grenoble, France

**Correspondence:** Sanjeev K. Jha (sanjeevj@iiserb.ac.in)

**Abstract.** Extreme precipitation events (EPEs) and flash floods incur huge damage to life and property in urban cities. Precipitation forecasts help predict extreme events; however, they have limitations in anticipating the impacts of extreme events. Impact-based forecasts (IBFs), when integrated with information of hazard, exposure and vulnerability, can anticipate the impacts and suggest emergency decisions. In this study, we present a serious game experiment, called the INSPIRE game, that

evaluates the roles of hazards, exposure, and vulnerability in a flash flood situation triggered by EPE. Participants make decisions in two rounds based on the extreme precipitation and flood that occurred over Mumbai on 26 July, 2005. In the first round, participants make decisions for the forthcoming EPE scheduled for later in the afternoon. In the second round, they make decisions for the compound events of extreme precipitation, river flood and high tide. Decisions are collected from 123 participants, predominantly Researchers, PhDs and Masters students. Results show that participant's use of information to

make decisions was based on the severity of the situation. A larger proportion of participants used precipitation forecast and exposure to make correct decisions in the first round, while used precipitation forecast and vulnerability in the second. Higher levels of education and research experience enabled participants to discriminate between the severity of the event and use the appropriate information set presented to them. Additionally, between the choice of qualitative and quantitative information of rainfall, 64% of the participants preferred qualitative over quantitative. Finally, we discuss the relevance and potential of

vulnerability integration in IBFs using inferences derived through the serious game.

## 1 Introduction

The accuracy and precision of Quantitative Precipitation Forecasts (QPFs) has undergone remarkable improvements in recent decades, owing to the advancement in computing technology and high-resolution data assimilation techniques (Kirkwood et al., 2021; Samal et al., 2023; Singhal et al., 2023). Today, QPFs are available with high spatial and temporal resolutions and large

areal extent. Availability of QPFs with lead times of up to 15 days have enabled the timely forecasting of hydrological extremes such as extreme precipitation and flash floods, reasonably well (Ahlgrimm et al., 2016). Recently, aplications of precipitation nowcasting have emerged which aim at high-resolution forecasting of rainfall, a couple of hours into the future, keeping in mind the socio-economic needs and local decision-making (Ravuri et al., 2021; Ballard et al., 2016; Laura Poletti et al., 2019).





Despite the increasing availability and performance of QPFs across the globe, loss of lives and economic damage have
continued to rise (Nanditha and Mishra, 2021; Lala et al., 2021; Singhal et al., 2022). Three main reasons may be advanced
as to why the improvements in QPFs have not necessarily led to better mitigating the losses of lives and property. First,
improvements in the QPFs, themselves, are still lacking quality to accurately predict the magnitude, intensity and duration of
extreme hazards (EPE or flash flood). Second, early warning systems have generally used QPFs to focus on hazards rather
than on their impacts at local scale. Third, the obtained hazard information is not well integrated with the information of local
exposure and vulnerability. The lack of integration of the vulnerability information may not provide a clear and comprehensive
understanding of preventive actions to the local public. Hence, there is a need for not just 'forecasts', but 'Impact-based
Forecasts (IBF)' informing the local public about 'What the weather will do' rather than just 'What the weather will be'
(Hemingway and Robbins, 2020; Kaltenberger et al., 2020).

A well-informed decision-making in IBF requires two types of information to assess risks: (a) information regarding the
hazard and (b) information regarding the area-specific vulnerability. Information regarding the hazard, such as its magnitude,
frequency, temporal duration and spatial extent, is available from the QPFs and is well documented in the literature (Papa-
giannaki et al., 2015; Coughlan De Perez et al., 2015; Robbins and Titley, 2018). However, human actions and interventions
produce vulnerability and, in particular, exposure, which are as important as the hazards themselves. Vulnerability is defined
as "the conditions determined by physical, social, economic and environmental factors that increase the susceptibility of an
individual, a community, assets or systems to the impacts of a particular hazard" (UNDRR, 2017). The information on vul-
nerability is crucial to provide guidance for effective adaptation planning and informed decision-making processes (Næss
et al., 2006; Parker et al., 2019; Singhal and Jha, 2021). On the other hand, exposure is defined as "the situation of people,
infrastructure, housing, production capacities and other tangible human assets located in hazard-prone areas" (UNDRR, 2017).
Decision-making in IBF should be an integrated framework involving information on hazards (single or compound), exposure
and vulnerability to enable decision-makers to take timely mitigating actions (Kox et al., 2018).

Serious games – games used for purposes other than entertainment – are a potential tool to train, improve and test decision-
making processes in a controlled environment (Rusca et al., 2012; Aubert et al., 2018). One objective of game-playing is
to transfer the lessons learnt to real-world decision-making (Geurts et al., 2007). Mayer (2009) defined serious games as
"experimental, rule-based, interactive environments, where players learn by taking actions and by experiencing their effects
through feedback mechanisms that are deliberately built into and around the game". Craven et al. (2017) consider serious
games as a space, free of the restrictions of the past systems, where scientists and policy-makers can come together. The game
presents a range of scenarios based on potential actions taken for various events, and each combination of selected actions for
an event has its own consequences (Aubert et al., 2019). Previous studies have noted that serious games are effective tools to
communicate advanced scientific concepts in fields such as climate adaptation, water resource management, urban planning
and disaster management (Arnal et al., 2016; Rumore et al., 2016; Flood et al., 2018; Crochemore et al., 2021). However, in
the context of hydrological hazards such as EPEs and floods, most of the serious games have focused on the understanding and
communication of the hazard information rather than on the role of exposure and vulnerability in the decision-making process
(Crochemore et al., 2016; Terti et al., 2019).





In this study, we present a serious game experiment which assesses the interplay between hazard, exposure and vulnerability in a situation of flash flood triggered by extreme rainfall. The 'INSPIRE flood' game is here designed based on the dramatic flood of 2005 in Mumbai; all values for hazard and vulnerability provided in the game being derived from observations of that particular event and area. Results from 123 answers collected during 6 game sessions are analyzed to explore how to best combine and communicate hazard and vulnerability information for emergency decision-making in IBF. We argue that, with hydrometeorological advancements, humans may be able to improve their capacity to forecast the magnitude, temporal duration and spatial extent of EPEs in the near future; however, this may not guarantee fewer impacts from such EPEs, compared to what we are witnessing today. One of the ways of mitigating such impacts is by making the socio-environmental systems less vulnerable (i.e. less exposed, less sensitive and more adaptable) to EPEs. Hence, we aim to show that vulnerability and its underlying components need to be included in IBFs and decision-making protocols. We test combinations of extreme rainfall, exposure and vulnerability to understand how the components of risk and their communication (i) alter decisions, (ii) influence confidence in decisions, and (iii) are perceived in support of flood-based decision-making.

## 2 Study Area and description of the extreme precipitation event

The game is inspired by the dramatic EPE and the consequent flood of 26-27 July 2005 in Mumbai, the financial capital of India. The city is located on the west coast of India surrounded by the Arabian Sea on all sides as shown in Figure 1(a). It is divided into 24 municipal wards (local administrative units) covering an area of 603 km2 (233 sq. mi) with a population of approximately 20 million. Among the four rivers flowing through the city, the Mithi river is the most prominent one acting as a natural drainage channel and carrying excess water during the monsoon season, i.e., from June to September (JJAS) out of the city towards the Arabian Sea.

On 26-27 July 2005, the city received an unprecedented precipitation of 944 mm over 24 hours with 190.3 mm rainfall in a single hour as observed by the Santa Cruz observatory (Jenamani et al., 2006). The annual average precipitation for the city is around 2000 mm. Over 60% of the city was flooded with large variability across wards (Figure 1b), mainly due to the extreme precipitation, and partly due to insufficient drainage systems, and a high tide of 4.48 meters which did not allow the flood water from the Mithi river (Figure 1c) to drain out to the sea (Chitale et al., 2006). As a result, the transportation and communication systems collapsed: the airport was closed, major roads were submerged, intercity trains were halted, and electricity was cut-off in many parts of the city. About 1200 people lost their lives either directly or indirectly because of the torrential rainfall and the subsequent deluge-related diseases (Gupta and Nair, 2011) and an economic damage of around 2 billion USD was incurred.

## 3 The INSPIRE game

The aim of the 'INSPIRE' flood game is to test out different ways of communicating jointly the hazard and vulnerability information for emergency decision-making. To this end, we test different joint visualizations of extreme rainfall, exposure and vulnerability information in a decision-making process resembling that of the 2005 Mumbai flood.



## 3.1 Game area

The wards 'K/W', 'H/E' and 'A' (see Figure 1a and 1c) were selected for the setting of the game so as to represent three different combinations of hazard and vulnerability levels. Wards 'K/W' (55%) and 'H/E' (90%) were the most affected wards in the event. Ward 'A' was the least affected ward and serves as a control-ward in the game as players should not make drastic decisions there. In addition, the three wards bring out different flooding drivers: ward 'K/W' was primarily flooded due to extreme precipitation, while 'H/E' was impacted by the combination of extreme precipitation and the overflowing of the Mithi river. Lastly, the wards represent varying socio-economic conditions, from huge slum population (ward 'H/E') to flourishing tourist hotspots (ward 'A') leading to different levels of vulnerability and exposure.

In the game and hereafter, the selected wards 'K/W', 'H/E' and 'A' are respectively referred to as towns 'Alpha', 'Beta' and 'Gamma' (Figure 1d). They represent parts of the fictitious INSPIRE city which was created for the game. The geographical and socio-economic characteristics of these three towns mimic that of the three corresponding wards of Mumbai. Participants make decisions for each of these three towns.

## 3.2 Game session unfolding

### 3.2.1 Introduction of the game

Each participant is assigned the role of 'Flood Risk Manager (FRM)' for an adapted city called 'INSPIRE' (Figure 1d). The main goal of each participant is to make the best possible emergency decisions to minimize the impact of the extreme precipitation and flood. The game moderator first informs the players of the area they will have to manage, with relevant geographical and socio-economic characteristics of the city, and of the decisions they will have to make. Each participant undertakes the game experiment independently and is given a worksheet to mark the decision choices: either to take an action, or not to take it (Yes/No decisions). The game moderator ends the introduction round by informing the participants of the risk of an upcoming extreme rainfall event across INSPIRE which may lead to flash floods.

### 3.2.2 Game rounds

The game is then set in two rounds corresponding to successive times in the progression of the event. At the beginning of each round, a crisis unit provides participants with field information (water level of the lake and river, sea tide height, prevailing ground situation and possible future developments) from the Meteorological Department, Department of Town Planning, Department of River Management, Department of Coast Management and the media cell (Figure 2). The participants are then provided with town-scale information of vulnerability, forecast accumulated rainfall and exposure in random order (Figure 2). Based on the received information, the participants decide which decisions need to be implemented. On an average, the participants required 22-25 minutes to play the two game rounds.





### 3.2.3 End survey

After the game rounds, participants are requested to complete a survey (Section D; Figure A1; Appendix A) to reflect on their perception of the provided combinations of information (i.e., most preferred combination, usefulness of exposure and vulnerability, qualitative or quantitative information of rainfall forecast etc.). Prescribed space for any feedback in the form of comments and suggestions were provided at the bottom of Section D.

### 3.2.4 Game scoring

For each decision, the participant is awarded 'approval points' (+10 and -10 for each correct and incorrect decision respectively). There are a total of 360 potential points available in each of the two rounds, resulting in a total of 720 potential points. At the end of the game, participants are given the correct and incorrect decisions, to calculate their score for each information set, and their total score. Correct and incorrect decision choices are defined by a game-play by decision-makers in Mumbai who often make real decisions in the wake of any extreme precipitation or flood event. The worksheets of those participants

are excluded from the analysis who make more decisions for Gamma (control ward), than Alpha or Beta in the first round, due to their lack of understanding of the flood situation or the rules of the game.

### 3.2.5 Debrief

After the game, a debriefing session of 10-15 minutes is convened to initiate and encourage discussion among the participants about the game experiments, its structure, usefulness of the information provided to make informed decisions and the approach

towards their choice of decisions. Finally, the actual backdrop of the game, the Mumbai flood, is revealed to the participants.

### 3.3 Information provided to players

During the game, the participants are provided with town-scale information of 4-hourly accumulated forecast rainfall, flood-prone population density, exposure and vulnerability based on which they make decisions (see Figure 2). The rainfall forecast is provided in both qualitative and quantitative forms in the game, to experiment different visualizations of rainfall informa-

140 tion. The information of flood-prone population density is provided quantitatively while exposure and vulnerability are only provided qualitatively.

The accumulated rainfall forecast, used in the game, is a slight modification of the actual rainfall observed during the event on 26-27 July, 2005. The modification ensures that the rainfall in the three selected towns is contrasted. The actual forecast issued for the event is not used in the game because it highly underestimates the actual rainfall, rendering it unreliable for

representing the actual developments in the event. The flood-prone population density (per sq. km) is calculated for the three selected wards as the total ward population multiplied by the percent area flooded during the 2005 event.

The information of exposure and vulnerability is statistically calculated using a set of indicators. Both exposure and vulnerability are calculated for each of the 24 wards of Mumbai. The values of the three selected wards (Alpha, Beta and Gamma) are then provided as qualitative color-codes to the participants during the game; the color-codes vary in four severity levels (I





to IV; denoting low, moderate, high and extreme) each corresponding to a quartile range, e.g. the 25% of the wards with the highest vulnerability are assigned a severity level IV (extreme).

### 3.4 Vulnerability and Exposure analysis

In this study, we consider vulnerability to be divided into three components- exposure, sensitivity and adaptive capacity (Gallopín, 2006). The values of each of the components and the overall vulnerability were calculated for all 24 wards prior to ward selection. The first step is to collect and identify relevant data (as 'indicators') and categorize them under the components of exposure, sensitivity and adaptive capacity. Each indicator is then standardized based on the actual area of the ward which was under flood during the 2005 event. For instance, close to 55% of Alpha's total area was flooded in the 2005 event, which implies that each indicator value of Alpha is considered to be 55% of its actual value in this study. This standardization is done to make sure that the vulnerability (and exposure) of each ward is calculated close to the reality of the 2005 flood event. Subsequently, each indicator is normalized, weighted and aggregated to form the vulnerability index followed by their four-level categorization.

#### 3.4.1 Rationale for indicators

Selection of appropriate indicators is essential for the accurate assessment of vulnerability. Several indicators may directly or indirectly influence the cause or impact of floods which can be used to assess the vulnerability of an area. Such indicators were obtained from publicly available data from government agencies (MCGM, 2022) and were selected to best define the flood vulnerability of wards of Mumbai. Table 1 lists all the selected indicators for this study along with their description and relationship with vulnerability.

#### 3.4.2 Normalization of indicators

The indicators are expressed in different units which require normalization before comparison. We use the maxima-minima method to normalize the indicators in this study (Singhal and Jha, 2021) as shown in Eq. (1).

$$X_{ij} = \frac{V_{ij} - MinV_{im}}{MaxV_{im} - MinV_{im}} \tag{1}$$

The normalized values are between 0 and 1, i represents the selected indicator, j is the selected ward, m is the total number of wards. $V_{ij}$ is the value of the $i^{th}$ indicator. $minV_{im}$ and $maxV_{im}$ are the minimum and maximum values of the $i^{th}$ indicator over the study area, respectively.

#### 3.4.3 Weighting of indicators

In this study, we assign weights to the indicators using the inverse variance method proposed by Iyengar and Sudarshan (1982). The method has been widely used in several vulnerability studies (Murthy et al., 2015; Omerkhil et al., 2020). The two-step equation is shown in Eq. (2) and Eq. (3).





$$W_i = \frac{k}{\sqrt{var_j X_{ij}}} \tag{2}$$

where the variance is calculated for a given indicator i and all wards j. k is a constant such that

$$k = \left[ \sum_{i=1}^{n} \frac{1}{\sqrt{var_j(X_{ij})}} \right] \tag{3}$$

$W_i$ is the weight of the indicator i (between 0 and 1) and the sum of all the assigned weights is equal to 1.

### 3.4.4    Exposure, sensitivity, adaptive capacity and vulnerability

The sub-indices of exposure (E), sensitivity (S) and adaptive capacity (AC) are calculated using the normalized values and
weights of the corresponding indicators (Balaganesh et al., 2020) using Eq. (4).

$$subindex = \frac{\sum_{i=1}^{n} W_i X_{ij}}{\sum_{i=1}^{n} W_i} \tag{4}$$

Lastly, we use the additive (averaging) approach to calculate the flood vulnerability index (VI). The approach aggregates the
sub-indices of exposure, sensitivity and adaptive capacity as shown in Eq. (5).

$$VI = \frac{E + S + (1 - AC)}{3} \tag{5}$$

### 3.4.5    Categorization of indices

The obtained indices (exposure, sensitivity, adaptive capacity and vulnerability) are categorized into four levels using the
approach proposed by Iyengar and Sudarshan (1982) as shown in Eq. (6). Each level (low, moderate, high and extreme)
is assigned 25% probability of occurrence based on the beta distribution which is generally appropriate to classify positive
random variable (Murthy et al., 2015).

$$f(z) = \frac{z^{a-1}(1-z)^{b-1}}{B(a,b)}, \quad 0 < z < 1 \text{ and } a, b > 0 \tag{6}$$

Where B (a, b) is referred to as the beta function defined by Eq. (7).

$$B(a,b) = \int_0^1 z^{a-1}(1-z)^{b-1} dz \tag{7}$$

The maximum likelihood approach is used to estimate the parameters a and b of beta distribution.



## 3.5 The experimental design

In the game, decisions are made in two rounds based on the two waves of extreme precipitation and flood that actually occurred over Mumbai on 26 July, 2005. The first round of decisions (R1) are made at 12:00 (noon) keeping in mind the forthcoming extreme precipitation event scheduled for later in the afternoon. The second round of decisions (R2) are made at 16:00 for the compound events of extreme precipitation, river flood and high tide later in the evening. Both rounds comprise three information sets each. The details of the rounds and experiments are presented in Table 2. The main aim of the experiments is

to test different combinations of the hazard (extreme precipitation) and vulnerability information and identify the best suited combination for emergency decision-making. The participants make decisions in both rounds based on the received information of extreme precipitation, exposure and vulnerability. In total, there are two rounds and three information sets which makes the total set of experiments equal to six. Hereafter, a combination of round and information sets is denoted Rm_En where m = 1, 2 and n = 1, 2, 3. For instance, R1_E1 represents the first round of decisions informed with the first information set. The three

information sets in each round were delivered in three different orders depending on the game session: E1-E2-E3, E3-E2-E1 and E2-E3-E1. This ensured that no combination of information had an advantage due to its placement as the first or last and that results are independent from the order in which the information sets were delivered.

In both R1 and R2, three different combinations of hazard and vulnerability information are provided to the players. The objective of the first round is to familiarize the participants with the different combinations of information, and more importantly,

to understand their choice of combination when making emergency-decisions in the wake of an extreme precipitation event. The second round is played to understand the choice of information made by the participants to achieve higher scores during compound events.

Specifically, in R1_E1 and R2_E1, the participants are provided with a quantitative forecast of the 4-hour accumulated rainfall along with the flood-prone population density for Alpha, Beta and Gamma. In R1_E2 and R2_E2, the participants

receive qualitative information (as color-codes) of rainfall forecast and exposure, while in R1_E3 and R2_E3, they are provided qualitative information of vulnerability along with the information of rainfall forecast (see Table 2). With these six experiments, the main aim is to identify the best possible combination of extreme rainfall and vulnerability information for emergency decision-making. Alongside, two other research questions are also addressed: (i) is the quantitative information of rainfall forecast and a single important indicator of flood prone population density sufficient for making optimal decisions during flash

floods? (ii) is it enough to consider exposure instead of vulnerability to improve emergency decision-making?

During each of the six experiments, the moderator provides the hazard and vulnerability information to the participant. The participant's role is to assess the provided information and accordingly, mark the decisions in a worksheet (Figure A1; Appendix A). After the decisions are made for each experiment, the participant also highlights the confidence in the decisions they made.





## 3.6 Game sessions and participants

A total of 123 worksheets were collected through five distinct presentations of the game (Table B1; Appendix B). All sessions were conducted in academic and research institutions and answers were collected on paper sheets. Participants start by filling in their profile (positions, field of expertise and self-rated knowledge) as shown in Figure 3.

The distribution of participants by position, field of expertise and self-rated knowledge in their field is shown in Figure 3. The majority of participants identify as senior-level students, PhDs (48%) and Masters (24%), followed by researchers (22%; Figure 3a). The category of researchers encompasses professors, scientists, and other academic positions related to science. Most participants (89%) identify their field of expertise as 'natural science', while the remaining participants identify as 'socio-economic science' (Figure 3b). Moreover, we asked the participants, "How much would you rate your knowledge in your field on a scale of 0 to 10?". Around 70% of the participants rated their knowledge levels between 4 and 7, while 27% rated themselves between 8 and 10 (Figure 3c). The remaining 3% considered their knowledge between 0 and 3.

## 4 Results

### 4.1 Overview of the decision-making

Twelve participants (approximately 10% of 123) had a clear misunderstanding of the game context as they made a greater number of decisions for the town of Gamma, than they did for Alpha or Beta in the first round. Based on the town-specific information provided to players, Gamma should have been considered the least affected town in the game. In total, 111 worksheets are analyzed.

The total possible score in the game (720), is divided into four quartiles. Around 14% of the participants scored in the first quartile (0 to 180), 62% in the second quartile (181 to 360), 24% in the third quartile (361 to 540), while one participant scored in the fourth quartile (541 to 720). The low scores of participants in the first quartile (lowest scores) can be attributed to them adopting a risk-prone strategy. The presence of only one participant in the last quartile indicates that the decision-making process in the game was not overly straightforward. Participants in the third quartile probably had a better understanding of the game, as they consistently made a greater number of correct decisions across experiments, regardless of the information provided to them.

Results show that the participants were better prepared to make decisions in the second round (Figure 4). Around 80% of the participants (89) scored higher in the second round than in the first (22). Seven participants scored the same in both rounds. Figure 4 (a-c) presents the cumulative distribution of scores in each of the three experiments from both rounds. Additionally, Kolmogorov-Smirnov (KS) tests were performed indicating that the distribution pairs are statistically different at the 5% level confidence. Participants scored higher in the second game round regardless of the experiment, but especially in E3 when they were provided with the combination of qualitative rainfall forecast and vulnerability. Consistently higher scores in the second round suggest that (a) there might have been a learning effect throughout the game, with better decisions being taken in the second round as participants became more familiar with the rules and the available information, or (b) the severity of the





forecast event might have played a role as a higher severity in the second game round led to more actions taken by participants, leading to higher scores.

## 4.2 The role of exposure and vulnerability in emergency decision-making

Figure 5 presents the cumulative distribution of the scores of the participants for each information set in the first (a) and second rounds (b). Here, our aim is to examine (i) if the different provided information led to different scores, and hence decisions, and, (ii) whether the information of exposure and vulnerability helped the participants in making correct emergency decisions. In the first round (R1), 43% of participants achieved higher scores when rainfall forecast and exposure (E2) information was given to them. In the second round, the highest percentage of participants (47%) scored the highest when the information of rainfall

forecast and vulnerability (E3) was provided. Figure 5a shows less distance between the curves of R1_E1 and R1_E2 than between R1_E1 and R1_E3, with R1_E2 displaying higher scores overall and R1_E3 displaying the lowest score distribution.

In the second round (Figure 5b), there is an evident change in the distribution of scores for all three information sets. All curves display higher median scores than in Round 1 as noted previously. R2_E1 and R2_E2 are located close together, and the tails of their distributions have moved significantly towards negative scores. This possibly implies that some participants

could not consistently make the correct decisions for the highly severe compound events in Round 2 based on the information provided in E1 or E2. The curve for E3, which consisted of the lowest scores in the first round, has considerably moved towards high positive scores in the second round. This indicates that most participants made a greater number of correct decisions when given the total vulnerability information, thus achieving higher scores. Overall, the participants scored well using the combination of rainfall forecast and exposure in the first round, and the rainfall forecast and vulnerability in the

second round, regardless of the order in which experiments were presented. The participants made different decisions based on the information of exposure alone and the full vulnerability information, and based on the severity of the crisis.

Further, we analyze the worksheets of the top 10 scorers in the game to understand which information they used. Among these participants, 8 out of 10 scored higher in the second round than in the first. In the first round, four participants reached higher scores with the information provided in experiment R1_E1 and four with R1_E2, while only two reached higher scores

with R1_E3. In the second round, five participants reached higher scores with the information in R2_E3, three with R2_E1 and two with R2_E2. The information supporting the high scores of the top 10 scorers is largely consistent with the results observed for the entire group of participants.

## 4.3 The role of education and knowledge in emergency decision-making

We select the participants with the following positions: 'Masters' (25 out of 111), 'PhDs' (55 out of 111) and 'Researchers'

(31 out of 111), to understand their preferred choice of information. These three positions were selected to understand how education level and research experience influence emergency decisions. We assume that the researchers have the highest level of education and experience, followed by PhDs and Masters. The participants were also asked to rate their knowledge in the field of expertise on a scale from 0 to 10. Based on the received information, we divide participants who rated their knowledge





between 0 and 6 (52 out of 111), and between 7 and 10 (59 out of 111). The inferences are drawn assuming that the participants were as honest as one can be while rating one's self.

We analyzed the scoring patterns depending on the education-level and self-rated knowledge. Results show that the combination of information used to achieve the best scores are largely similar to the overall trend which was observed for all 111 participants (Section 4.2). Moreover, we calculated the mean scores achieved by each of the three positions to infer if the level of education and experience played a role in achieving higher scores. Table 3a presents the mean score obtained by the three positions in the two rounds. Researchers have the highest mean score in the first round followed by the PhDs and Masters. A similar trend is observed in the second round with Researchers performing comparatively better than both PhDs and Masters suggesting that the level of education does play a role in decision-making. Further, Table 3b presents the mean scores based on their self-rated knowledge. Results show that the participants reach almost similar mean scores in the first round irrespective of their declared ratings. In the second round, the participants who rated themselves lower obtained higher mean scores (197) than the participants who rated themselves higher (179). This suggests that self-declared knowledge may not be a reliable indicator of a participant's decision-making ability.

### 4.4 The role of hazard-vulnerability context in emergency decision-making

In this section, we analyze the scores of the participants for each of the three selected towns in the study - Alpha, Beta and Gamma. In the first round, when the event was announced and starting, participants achieved the highest mean score for the town Alpha (47), followed by towns Gamma (36) and Beta (21). In the second round, when the severity of the event is announced and several drivers of flood come into play, the highest mean score is obtained for town Beta (67), followed by towns Alpha (49) and Gamma (45). In Section 4.2, results showed that participants primarily used the information of rainfall forecast + exposure to achieve higher scores in the first round, while rainfall forecast + vulnerability was mainly used in the second. This suggests that the information of vulnerability helps to make better decisions compared to the information of exposure. However, almost similar mean scores for Alpha in both the rounds indicate that contexts of ground-level hazards and vulnerabilities likely influenced decisions, just like information sets did.

Table 4 presents the level of hazard, exposure and vulnerability estimated for the three towns in the two rounds. Here, the levels of exposure and vulnerability in the towns vary with the rounds because the indicators used to calculate them are standardized based on the actual area of the ward which was flooded in 2005. Moreover, the cumulative distribution of each participant's score in the three towns are presented in Figure 6. Score distributions show that the participants used the information in E3 to make the best decisions in all the three towns, followed by E2 and E1, irrespective of the combination of hazard-vulnerability prevalent in the three towns. Greater differences between the three information sets are observed in Alpha and Beta, where the intensity of the hazard was the greatest (moderate to high in Alpha and high to extreme in Beta). This suggests that in these towns where high hazard severity required drastic measures (all actions taken), the type of adjacent vulnerability information played a discriminating role. On the opposite, in town Gamma, where the hazard severity was low to moderate, and where actions should have been taken parsimoniously, the vulnerability information did not necessarily help discriminate between actions to take.





### 4.5 Perceived usefulness of information compared to actual decision-making

In the final survey, we asked the participants a few questions on their perception of the usefulness of the different information
sets (see Figure A1; Appendix A). In this section, we first analyze the preferences of participants in terms of information sets,
and then assess if these preferences are consistent with the scores, they obtain for each information set.

#### 4.5.1 Participants prefer exposure and vulnerability information for emergency decision-making

In the final survey, we asked the participants about the usefulness of the exposure and vulnerability information in emergency
decision-making. We gave them four options to choose from - not useful, slightly useful, useful and very useful. Out of 111
participants, around 9% found the information slightly useful, 50% and 41% found them useful and very useful respectively,
while none found the information as not useful (Figure 7). This suggests that the information of exposure and vulnerability
helped participants in making correct decisions.

#### 4.5.2 Participants prefer qualitative rainfall information and exposure/vulnerability information for making decisions

We asked the participants in the final survey, "if you had only one kind of information for making decisions, which one would
you have preferred?" In response, 21 out of 111 participants picked the combination of quantitative rainfall forecast and flood
prone population density (E1) as the most useful to make decisions, and 29 out of 111 picked qualitative rainfall forecast
and exposure (E2) as the best information (Fig 8a). The highest number of participants (61 out of 111) chose the qualitative
rainfall forecast along with vulnerability (E3) as the most suitable to make decisions. We examine if the choices made by the
345 participants in the final survey are consistent with the scores they obtained in the game rounds. Results show that out of the 21
participants who consciously opted for E1, 23% achieved higher scores in R1_E1 (using E1) than in R1_E2 and R1_E3, and
19% achieved higher scores in R2_E1 than in R2_E2 or R2_E3 (Figure 8b). Therefore, a small percentage of the participants
who preferred E1 in the survey actually achieved high scores with E1. Moreover, among the 29 participants who opted for E2
in the final survey, 47% actually scored higher with E2 in the first round, however, the percentage dropped to 21% in the second
round. Out of 61 participants who perceived E3 to be the most suitable information, 49% of them achieved higher scores in
R2_E3 and 29% in R1_E3.

Results suggest that the participants' perceived usefulness of information and its actual usefulness do not match. The largest
difference is observed in the case of E1 where participants could not obtain higher scores in any of the two rounds, using the
information of their choice. Close to half of the participants who preferred E2 or E3 actually achieved higher scores with their
preferred information set in either the first or the second round. The participants who preferred E2 in the survey achieved higher
scores in the first round, while those who opted for E3 achieved the same only in the second round.





### 4.5.3 Participants find qualitative information of rainfall suitable for making decisions

When asked in the final survey about their preference between the qualitative and quantitative information of rainfall, 64 out of 111 participants responded in favor of qualitative, while 47 out of 111 opted for the quantitative form of rainfall (Fig 9a).
We compare these preferences with the scores they obtained during game play. Each player made a total of 72 decisions during the game in both rounds. Out of these 72 decisions, 24 were made using the quantitative information of rainfall, while the remaining 48 were made with the qualitative information. Then, we examine the total number of correct decisions made by the participants in their respective preferences. The 64 participants who preferred the qualitative information made on average 56% of correct decisions based on qualitative information in the first round, and 63% correct in the second (Fig 9b). For the
remaining 47 participants (who preferred the quantitative rainfall information in the end survey), 38% of the decisions were correct in the first-round while only 32% were correct in the second round (Fig 9c).

## 5 Discussions

### 5.1 The role of exposure and vulnerability in decision-making

One of the main research questions in this study is to ascertain which type of vulnerability information (between a population-
370 based single indicator, exposure and vulnerability) can best possibly combine with the extreme rainfall and flood information, for emergency decision-making in IBF. Most of the participants made the best decisions using the qualitative rainfall forecast and exposure information in the first game round (extreme precipitation round). Interestingly, when the information of vulnerability was provided along with rainfall in the first round, far less number of correct decisions were made. In the second round, the participants used the qualitative rainfall forecast and vulnerability information to make the best possible decisions
against compound events of extreme precipitation, river overflow and high tide. Vulnerability contained a larger scope of ground-level information than exposure. This implies that, in emergency situations, it is important to select the information that best suits the decisions to be made, as also pointed out by (Misra et al., 2020). Mere availability of a greater (and possibly more complicated) information can lead to information overload (van den Homberg et al., 2018), thus over-complicating the decision-making process.
One of the main challenges in the game was to make sure that the participants differentiated between the exposure-vulnerability information, and use them according to the situation of the game. Differentiating between the two information was tricky since they could only see the colors representing the severity of exposure/vulnerability, and not the actual indicators or values used to create them. The participants used different information choices in the two rounds of the game to make decisions which suggests that they could understand the difference between information. Also, it indicates that the information
of exposure and vulnerability have different roles in the decision-making process which is determined by the severity of the ground situation.





## 5.2  The role of education and knowledge in decision-making

Similar to Kim et al. (2018), results from the game show that higher education level and research experience leads to better emergency decision-making . Researchers obtained the highest mean score in both the rounds followed by the PhDs and Masters. Researchers, due to their higher education level, may have better understood the different hazard-vulnerability contexts of the two rounds. Higher research experience possibly allowed them to select the best information to make the decisions making them more consistent in decision-making across three towns. The masters made significantly more correct decisions in the second round than in the first, suggesting that they were able to adapt to the decision-making process.

Results from self-declared knowledge do not show a particular pattern in decision-making suggesting that it may not be a reliable indicator of a participant's decision-making ability. The possible reasons behind lack of a trend could be that (i) rating your own expertise in a research area is not simple, or (ii) a higher knowledge-level does not always correspond to better emergency decision-making abilities.

## 5.3  The role of hazard-vulnerability context in decision-making

The ground-level situation of hazard and vulnerability determine which information can better manage the severity of the situation. For instance, when the intensity of the hazard was the greatest (such as in Alpha and Beta), the information of vulnerability played a discriminating role in suggesting the participants which decisions to actually make. When the hazard severity was low (as in Gamma), the same vulnerability information could not always guide the participants in making decisions. This might suggest that vulnerability information helped confirm the drastic measures in extreme situations. In towns with low to moderate risks, a good understanding of actions and the level of risk-aversion of participants may have played a greater role than the format of the vulnerability information.

## 5.4  Perceived and actual usefulness of the presented information

Each participant had its own understanding of hazard, exposure and vulnerability, and upon playing the game, a personal perception and experience of these concepts in the game. We asked the participants in the final survey about their perceived usefulness of the exposure and vulnerability information in making decisions, almost all the participants (91%) found both the information useful to very useful. On the most preferred combination of information, the majority of participants selected rainfall forecast + vulnerability, however, only a small percentage of them (29% and 49% in the first and second round, respectively) could actually use it to make the correct decisions. Having said that, the perception of an information may not always match with the highest obtained score. Preference may not be based solely on the score outcome, but also on the ease of understanding the provided information (e.g. qualitative versus quantitative information), and on whether participants felt they had comprehensive enough information to make the right decisions (e.g. exposure versus vulnerability). Interestingly, a larger proportion of participants (58%) preferred the qualitative information of rainfall over the quantitative information in the survey. Results suggest that the participants who preferred the qualitative rainfall in the final survey were actually able to use it to make correct decisions during the game (56% and 63% in the first and second round, respectively). However, a significant





portion of participants who selected the quantitative rainfall in the end survey were unable to reach better decisions with the information.

## 5.5 Game limitations and prospects for future research

Although based on real events, the concepts involved in the INSPIRE flood game are wider-ranging than its storyline portraying a flash flood situation in an urban context. Flash flood was considered as the hazard in the game, however other natural or anthropogenic-induced hazards could also be included to assess the role of exposure and vulnerability in decision-making. Moreover, a single participant could take all four possible decisions in the game. Roles can be distributed among players to better understand the participant's thought-process behind each decision as shown by Terti et al. (2019). The role of rainfall and flood forecast, its reliability and behavior were not evaluated in the game which was more of a focus-point in other similar games (Arnal et al., 2016; Guido et al., 2023). It could be interesting (and a bit more complicated) to comprehend the decision-making process when all three factors, i.e., hazard, exposure and vulnerability show uncertainty. Moreover, since the game was played on worksheets with no real risk of damage to life and property, participants may have behaved as risk-prone and did not take sufficient actions, or as risk-averse, taking unnecessary actions. To avoid additional complexity, it was assumed that the participants had unlimited resources and personnels to make and implement the decisions. Inclusion of some constraints such as fixed number of volunteers for each decision, budgetary constraints or fictitious lives at stake could enhance the potential reality of the game. The storyline of this game was real, however, oversimplification of such drastic events may impact the transfer of overall outcome from the game to real-life decision-making, as opined by Aubert et al. (2019).

While in the present work only exposure as a component of vulnerability was included, future work may add other components such as sensitivity and adaptive capacity in the game to enhance the socio-economic understanding of an area. Addition of more such concepts would enable the participants to better anticipate the probable impacts, especially in regions which witness frequent floods (Weis et al., 2016). Moreover, future research could explore designing a two-step game in which the decisions made by the participants are validated by people and stakeholders for whom the decisions are actually taken, instead of the decision-makers. These stakeholders can be office goers, parents, local shop owners, transport-owners etc. who opine whether the decisions made by the participants are actionable to them or not. This will enhance the reliability of decisions that are made on worksheets during the game and will help in bridging the gap between in-game decision-making and its practical implementation on the ground. Finally, there is a need for better communication, and possibly greater simplification, of the exposure and vulnerability information provided to the participants before the start of the game. To address this, future works may focus on presenting the information differently, which may also include the actual indicators used to create the qualitative color-codes, as quantitative values or in the form of a warning matrix.

## 6 Conclusions

This paper presents a game experiment designed to simulate emergency decision-making in a situation of flash flood triggered by extreme rainfall, inspired by the actual event of 2005 in Mumbai, India. The game is designed to understand the roles of



hazard, exposure and vulnerability information in impact-based forecasting and their perceived usefulness by decision-makers for emergency management. It was played by a total of 123 participants in six distinct in-person sessions held in academic and research institutions across India. In the game, participants were sequentially provided with town-scale information of 4-hourly accumulated rainfall forecast, flood prone population density, exposure and vulnerability based on which they made decisions.

In the first round, participants made decisions in response to an extreme rainfall event, while in the second round they made the same decisions for compound events of extreme rainfall, flash flood and high tide.

During the game, participants were supposed to consider the given information in isolation and make fresh decisions each time, without having any influence of the previous information provided in the previous experiment or round. In the first round, a large proportion of the participants achieved higher scores using the combination of rainfall forecast and exposure. In

the second round, a majority of them obtained higher scores when the information of rainfall forecast and vulnerability was provided to them, regardless of the order of experiments.

The conclusions drawn from this game can help in further progress of impact-based forecasting of extreme hydro-climatic events. First, it is important to include the information of local exposure and vulnerability, along with the rainfall-flood forecast, while developing impact-based approaches or services. The information of exposure is particularly useful in making

preliminary decisions in the wake of an extreme hydrological event, while vulnerability is effective, if and when the ground situation has worsened over a period of time. During the feedback session, several participants, while acknowledging the relevance of exposure and vulnerability, suggested providing a broader overview of the concepts before the start of the game. Second, the participants demonstrated significant improvement in decision-making in the second round of the game. One of the clear reasons for this improvement is that they developed a greater understanding of the decision-making process in the

first round. After the game, few participants in every session wished to play the first round again. When reminded to them that they already know the correct decisions, they asked to play a different game, as they believed they could make better decisions this time around. This strongly indicates that there is a need for a greater focus on training and developing decision-makers by putting them in diverse scenarios, presenting them with different information and working on their risk-taking behavior to build a comprehensive decision-making expertise. Finally, the game is a simplified representation of reality having no ability

or intention to replicate the actual event which occurred over Mumbai in 2005. The main purpose is to experience, investigate and discuss the challenges of decision-making in such emergency situations and convey the possible solutions to real-world impact-based forecasting as much as possible. The prospect of saving your town from flooding, collecting negative points for incorrect decisions, the possibility of getting fired from their jobs, and the joy of saving your town in the end added a light touch to the game and created an inclusive environment to discuss the usefulness of different information related to emergency

decisions for impact-based forecasting. The developed decision-making framework can be useful to forecasters, meteorological departments, urban planners, policy-makers and disaster response authorities to make well-informed decisions and generate effective impact-based forecasts and early warnings.

*Data availability.* The data that support the findings of this study are available from the corresponding author upon reasonable request.



*Author contributions.* **A Singhal:** Conceptualization, Formal analysis, Writing – original draft, Investigation, Visualization and Funding
acquisition. **L Crochemore:** Methodology, Conceptualization, Visualization, Writing and editing manuscript; Validation and Supervision. **I
Ruin:** Methodology, Visualization; Conceptualization, Writing and editing manuscript; Validation. **SK Jha:** Methodology, Conceptualization, Visualization, Validation, Writing and editing manuscript; Supervision

*Competing interests.* The authors declare they have no conflict of interest.

*Acknowledgements.* We thank all the participants who took part in this experiment and the volunteers who played the game to test the
490 game and fix any obvious shortcoming. We would like to thank the host institutes Central University of South Bihar, Indian Institute of
Science Education and Research, Bhopal, Indian Institute of Tropical Meteorology, Pune and Municipal Corporation of Greater Mumbai
for providing us the necessary facilities to conduct the game sessions. We especially acknowledge the support provided by the Disaster
Management Authority, Municipal Corporation of Greater Mumbai in providing us all the relevant data and information about the situation
of floods in Mumbai. We also thank the representative for playing the game and providing us with the optimal decisions based on which
the decisions of the participants were evaluated. The resources and support provided by the Institute of Environmental Geosciences at the
Université Grenoble Alpes, where the concept of the game was actually developed, is also acknowledged.

Akshay Singhal acknowledges the Raman-Charpak award by the Indo-French Centre for the Promotion of Advanced Research (IFC-
PAR/CEFIPRA; IFC/4141/RCF 2022/385) to carry out this research work. Akshay also thanks the Department of Science and Technology,
Government of India (DST/INSPIRE/03/ 2019/001343) (IF 190257) for financial support during the PhD under the DST-INSPIRE scheme.
Sanjeev Kumar Jha acknowledges the support of the Science and Engineering Research Board (SERB), project number CRG/2022/004006.





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





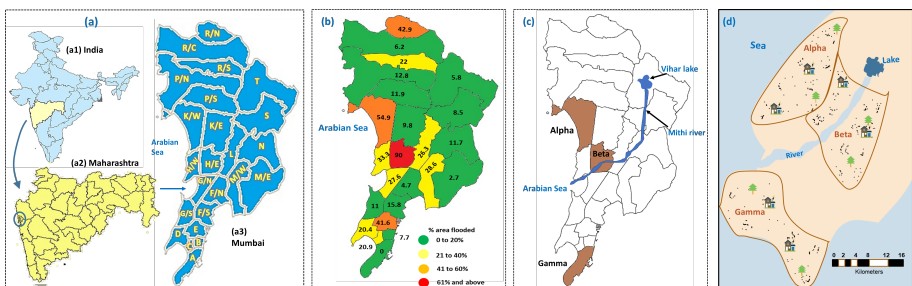

**Figure 1.** Maps of the study area: (a) Location of Mumbai in India; (b) Ward-wise percent area flooded during the extreme precipitation and flood event of 2005 (based on MCGM, 2006); (c) Wards selected for this study along with the flow direction of the Mithi river, originating from the Vihar lake; and (d) An illustrative map of the three selected wards named Alpha, Beta and Gamma in the fictitious city of INSPIRE.





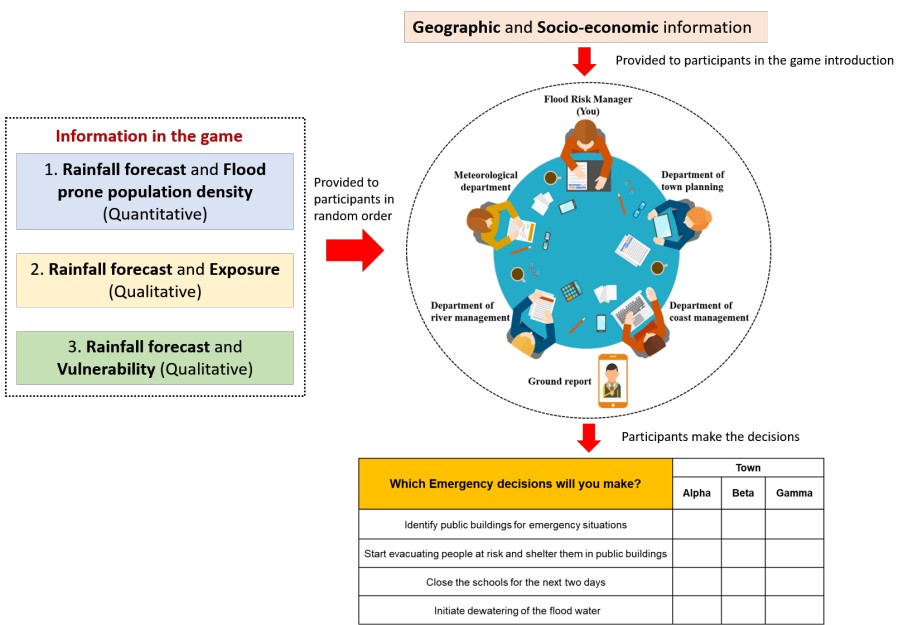

**Figure 2.** Different information provided to the participants to make the four emergency decisions for each of the towns Alpha, Beta and Gamma.





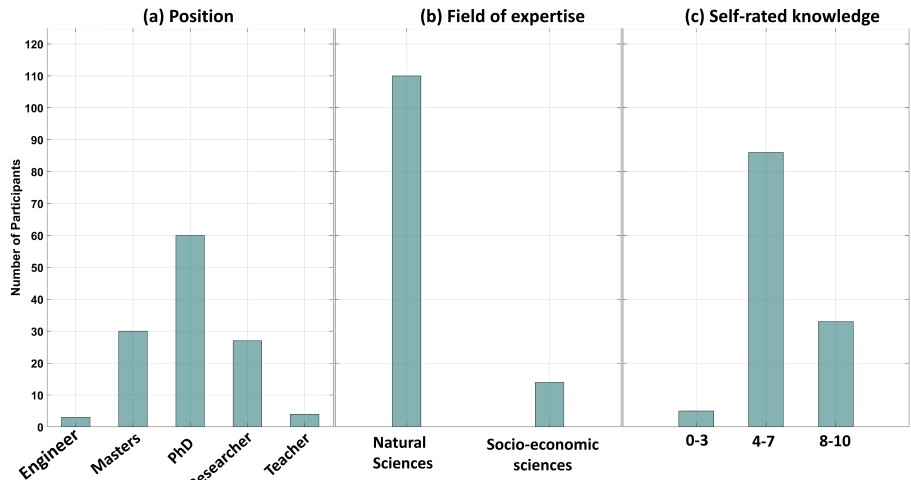

**Figure 3.** The information about the participants pertaining to their (a) position, (b) field of expertise and (c) self-rated knowledge in their field obtained from the worksheet.





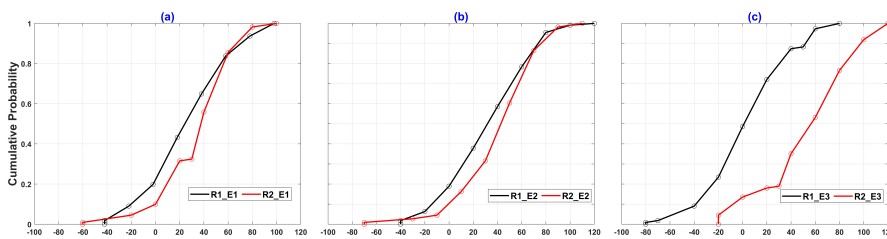

**Figure 4.** Cumulative distributions of the scores for information sets E1 (a), E2 (b), E3 (c) from Round 1 (no compound event, black line) and Round 2 (compound event, red line).



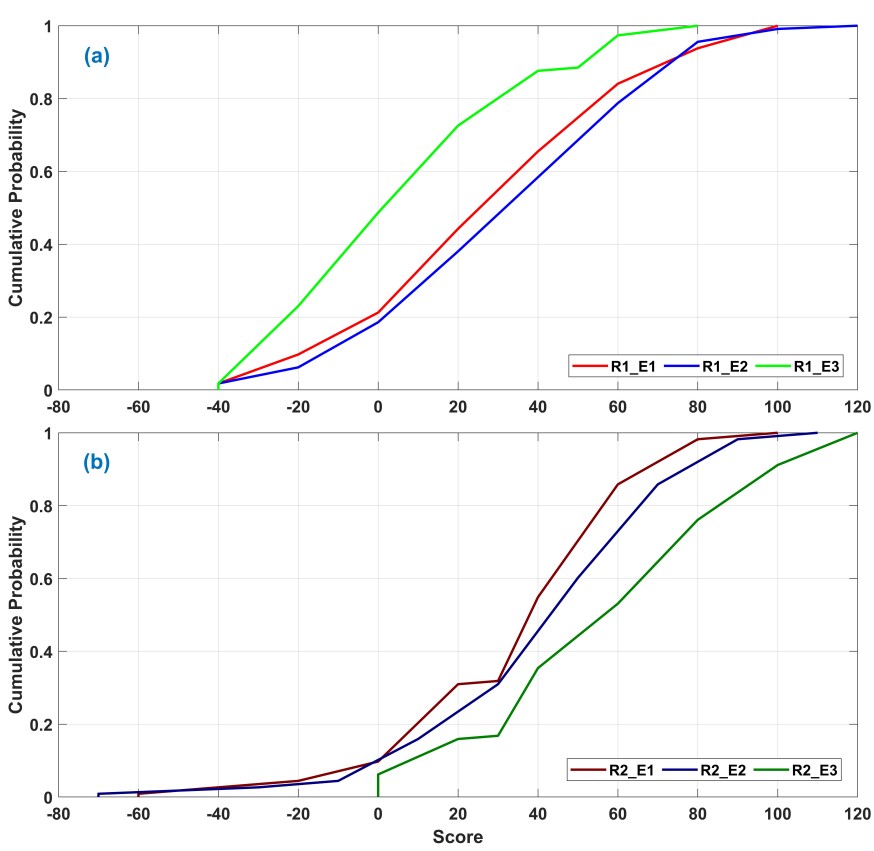

**Figure 5.** Cumulative distributions of the scores of all 111 participants depending on the information set (E1, E2, E3) (a) in the first round, R1 and (b) in the second round, R2.

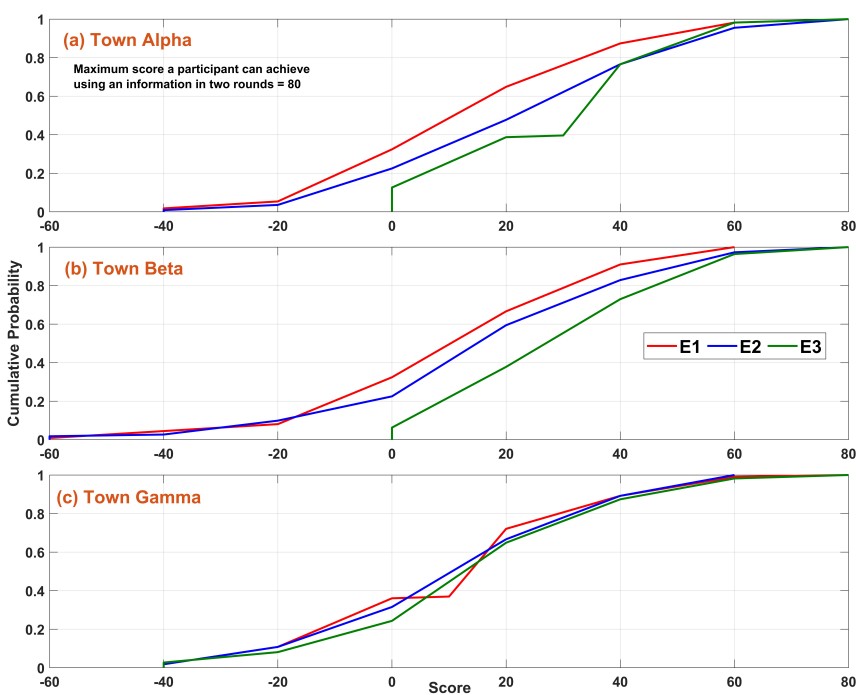

**Figure 6.** Cumulative distributions of the 111 participants' scores in the three towns of Alpha, Beta and Gamma depending upon the information E1, E2 and E3.



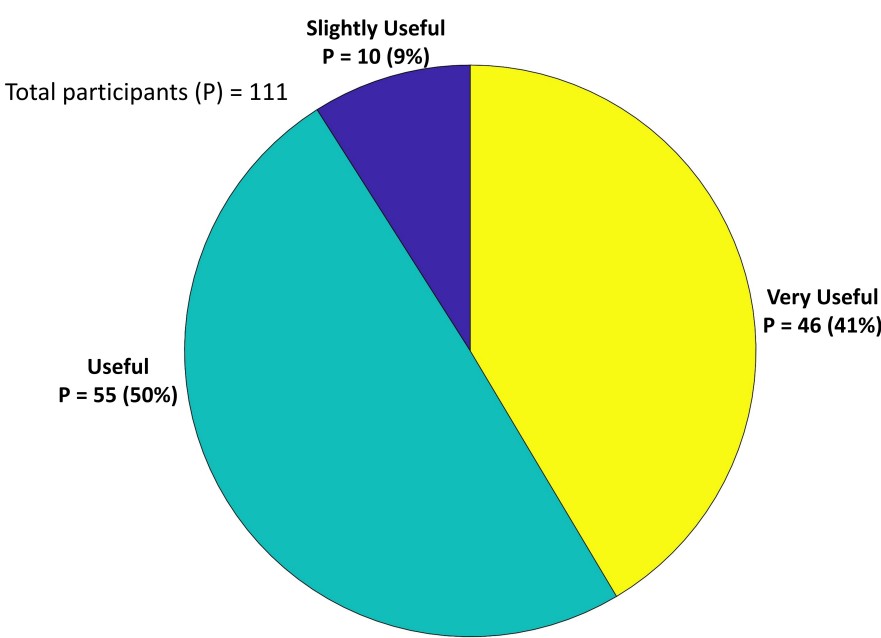

**Figure 7.** Participant's answers, in the final survey, regarding the usefulness of the exposure and vulnerability information in decision-making.





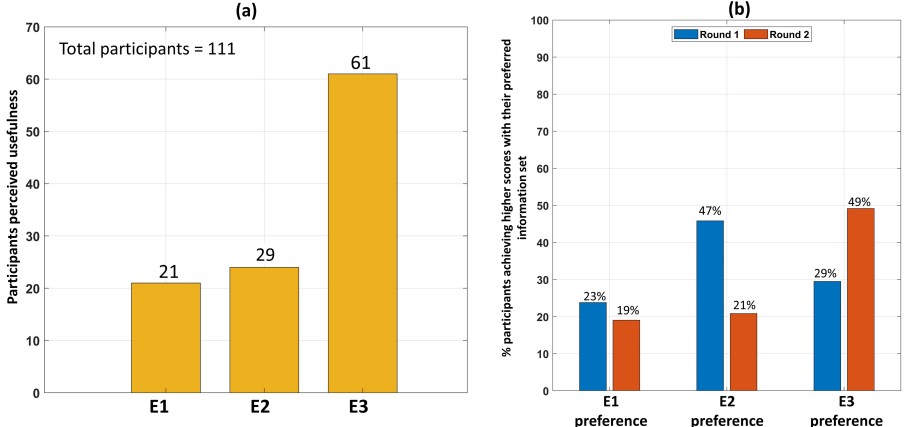

**Figure 8.** Description of (a) participants' preference for the different information sets (E1, E2 and E3) as declared in the final survey, and (b) percent of participants with a preference for each information set actually achieving a higher score with that preferred information set.



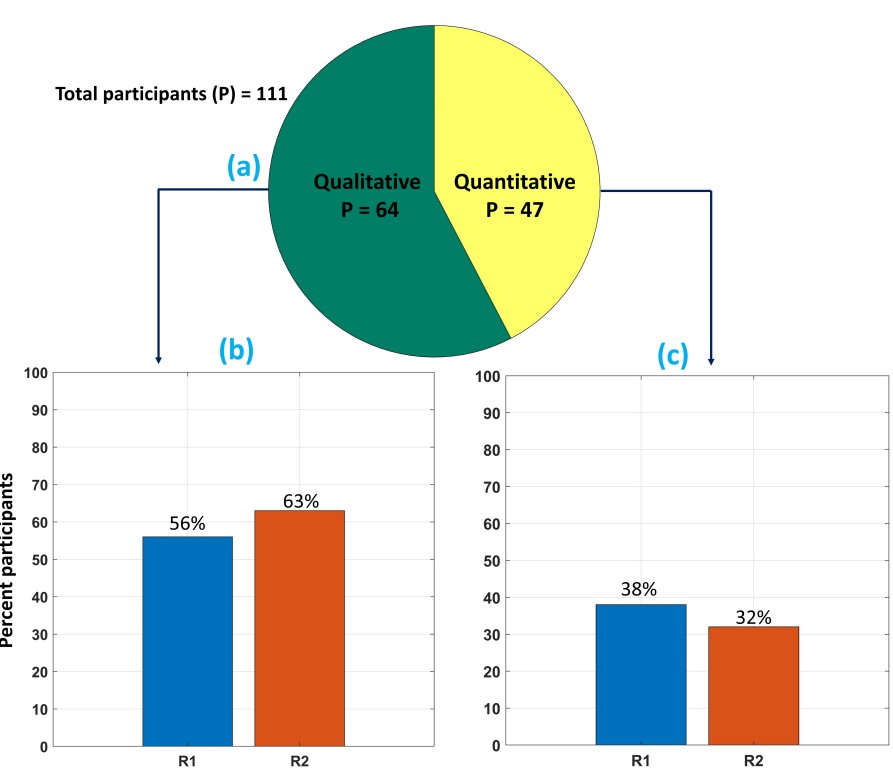

**Figure 9.** Description of (a) participants' preference, in the final survey, for the qualitative or quantitative information of rainfall forecast provided and (b) participants' actual performance during the two game rounds (R1 and R2) using qualitative rainfall information and (c) quantitative rainfall information.

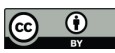



**Table 1.** The indicators selected for vulnerability assessment of the 24 wards. Here, (+) indicates that the vulnerability increases with an increase in the magnitude of the corresponding indicator and, (-) indicates that the vulnerability decreases with an increase in the magnitude of the corresponding indicator

| Component | Indicator | Description | Relationship with vulnerability |
|---|---|---|---|
| **Exposure** | Number of schools | Total number of schools in a ward | Positive |
| | Population density | Population living in per sq. km area | Positive |
| | Flood prone area (%) | Percent area flooded during the 2005 event | Positive |
| | Number of vulnerable settlements | Residential areas that are particularly prone to floods | Positive |
| | Homeless Population | No. of people living in temporary or inadequate arrangements | Positive |
| | Slum Population | No. of people living in densely populated areas with limited access to basic services | Positive |
| **Sensitivity** | Water Logging Spots | No. of designated areas where excess water accumulates due to poor drainage or heavy rainfall | Positive |
| | Total Illiterate population | No. of people who lack the ability to read and write | Positive |
| | Number Of Outfalls | No. of discharge points where stormwater is released into a natural water body | Positive |
| | Number of dilapidated buildings | Buildings that pose safety hazards to floods due to neglect or age | Positive |
| **Adaptive Capacity** | Number of roads | Number of paved roads designed for vehicular travel | Negative |
| | Length of roads (km) | Length of paved roads designed for vehicular travel | Negative |
| | Number of hospitals | No. of medical facilities equipped to provide healthcare facilities | Negative |
| | Emergency Assembling Points | Locations where individuals gather during a crisis | Negative |
| | Number of Dewatering Pumps | No. of machines used to remove excess water from flooded or waterlogged areas | Negative |





**Table 2.** Details of the experiments involved in the study.

| Round | Experiment | Hazard information | Vulnerability information | Form of vulnerability | Compound events |
|:---:|:---:|:---:|:---:|:---:|:---:|
| **R1** | R1_E1 | Rainfall forecast | Flood prone population | Quantitative | No |
| | R1_E2 | Rainfall forecast | Exposure level | Qualitative | No |
| | R1_E3 | Rainfall forecast | Vulnerability level | Qualitative | No |
| **R2** | R2_E1 | Rainfall forecast | Flood prone population | Quantitative | EPE; River flood; High tide |
| | R2_E2 | Rainfall forecast | Exposure level | Qualitative | EPE; River flood; High tide |
| | R2_E3 | Rainfall forecast | Vulnerability level | Qualitative | EPE; River flood; High tide |





**Table 3.** The mean and cumulative score achieved by the participants in the two rounds based on (a) their position-level and (b) their self-rated knowledge.

| (a) Position-level | Round 1 | Round 2 |
|---|---|---|
| **Master** | 78 | 173 |
| **PhD** | 109 | 189 |
| **Researcher** | 134 | 223 |
| **(b) Self-rated knowledge** | | |
| **0 to 6** | 109 | 197 |
| **7 to 10** | 111 | 179 |





**Table 4.** Levels of hazard, exposure and vulnerability calculated for the three towns in the two rounds. Here, * denotes the occurrence of compound events.

|  | Alpha | | | Beta | | | Gamma | | |
|---|---|---|---|---|---|---|---|---|---|
|  | $H_{level}$ | $E_{level}$ | $V_{level}$ | $H_{level}$ | $E_{level}$ | $V_{level}$ | $H_{level}$ | $E_{level}$ | $V_{level}$ |
| **Round 1** | Moderate | Extreme | Extreme | High | Moderate | Low | Low | Low | Moderate |
| **Round 2** | High | High | Extreme | Extreme | High* | Moderate | Moderate | Low | Low |





610 **Appendix A: Game worksheet**

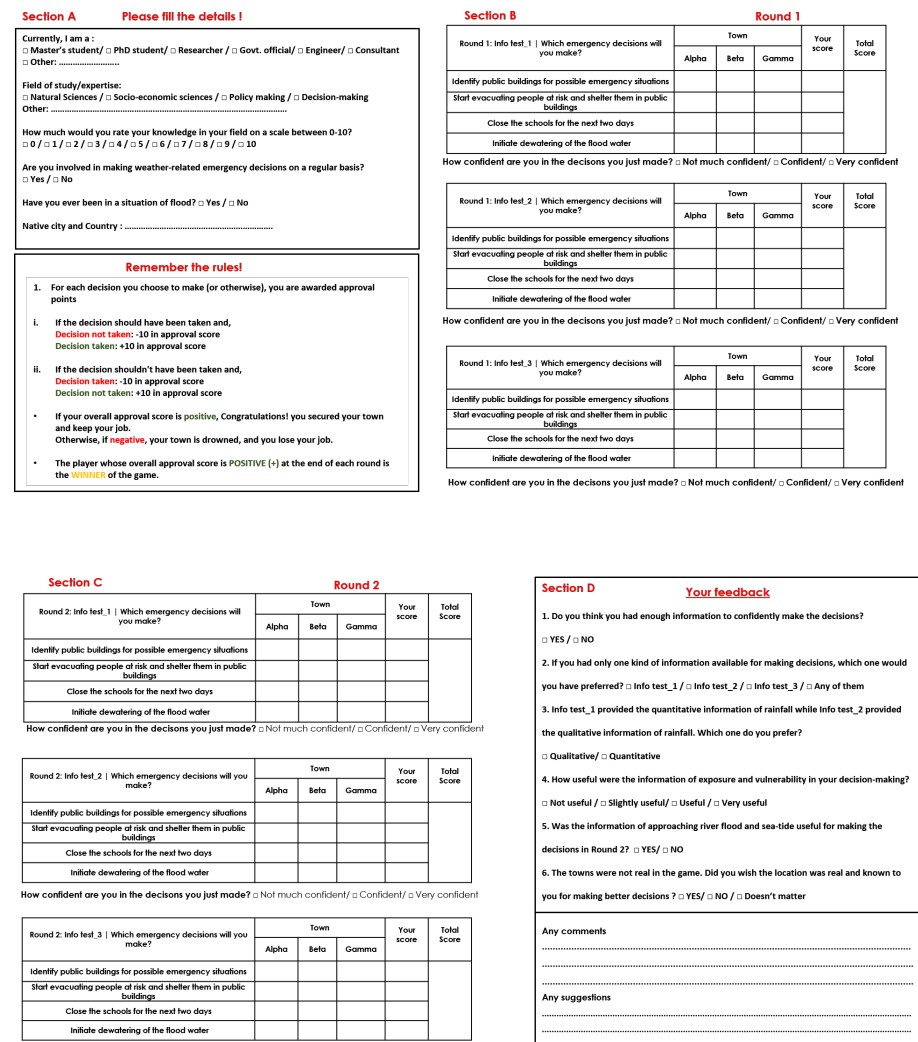

**Figure A1.** The worksheet (2 pages) distributed to the game participants in which they marked their decisions. The first page consists of Section A where the participants fill in some of their personal details, and Section B where they mark the decisions for the first round. The second page consists of Section C where the participants mark the decisions for the second round, and in Section D they submit their responses to a final survey.



**Appendix B: The sessions of the game**





**Table B1.** Details of the game sessions convened at different host institutes/organizations.

| | Host | Location | Worksheets collected | Participants | Length of the game (in mins) | Order of experiment |
|---|---|---|---|---|---|---|
| 1. | Central University of South Bihar | Gaya, Bihar | 30 | Master's and PhD students | 46 | E2- E3- E1 |
| 2. | Indian Institute of Science Education and Research | Bhopal, Madhya Pradesh | 10 | Master's and PhD students | 40 | E2- E3- E1 |
| 3. | Indian Institute of Science Education and Research | Bhopal, Madhya Pradesh | 36 | Master's and PhD students | 42 | E3- E2- E1 |
| 4. | Indian Institute of Tropical Meteorology | Pune, Maharashtra | 41 | Researchers, Scientists, Master's and PhD students | 43 | E1- E2- E3 |
| 5. | Municipal Corporation of Greater Mumbai | Mumbai, Maharashtra | 01 | Decision-maker | 36 | E1- E2- E3 |
| 6. | Others | Multiple | 06 | Teachers, Engineers | 40 | E3- E2- E1 |





## Appendix C: Optimal decisions

### Round 1

| Which emergency decisions will you make? | Town | | |
|---|---|---|---|
| | Alpha | Beta | Gamma |
| Identify public buildings for possible emergency situations | Y | Y | Y |
| Start evacuating people at risk and shelter them in public buildings | Y | Y | N |
| Close the schools for the next two days | Y | Y | Y |
| Initiate dewatering of the flood water | Y | Y | N |

### Round 2

| Which emergency decisions will you make? | Town | | |
|---|---|---|---|
| | Alpha | Beta | Gamma |
| Identify public buildings for possible emergency situations | Y | Y | Y |
| Start evacuating people at risk and shelter them in public buildings | Y | Y | N |
| Close the schools for the next two days | Y | Y | Y |
| Initiate dewatering of the flood water | Y | Y | Y |

**Figure C1.** The optimal decisions based on which the decisions of the participants were evaluated. The decisions were obtained from a representative of the MCGM, Mumbai.