# Peer review of "INSPIRE Game: Integration of vulnerability in impact-based forecasting of urban floods"

_Hydrology and Earth System Sciences, 2024_

## Author Response (AR1)

**Comments from reviewer 1 and their response**

**Overview:**

This paper reviews use of a 'serious game' in exploring how different types of information influences flood management decisions, here tested in the city of Mumbai, India. I comment here as someone who is familiar with urban flood management, but less familiar with use of 'serious games.' Overall, I think this is an important concept to explore how including information about exposure and vulnerability (rather than just the hazard itself) could improve decision making. However, as articulated below, I do think multiple aspects need to be clarified or explained more.

**General Comments:**

**Comment 1:** It seems like there could be some more description of previous work in this realm, including adding this helpful recent review of serious games and flood risk (https://wires.onlinelibrary.wiley.com/doi/full/10.1002/wat2.1589).

**Response:** Thank you for the suggestion. We have included the following description of previous work related to serious games in the Introduction Section of the revised manuscript at lines 65-71.

"Arnal et al. (2016) and Crochemore et al. (2016) designed serious games to better understand the perception and use of probabilistic forecasts in flood-related decision-making contexts. Terti et al. (2019) created a role-playing game called ANYCaRE simulating a crisis management unit to explore the value of modern impact-based weather forecasts on the decision-making process related to weather-risks in Europe. Sermet et al. (2020) developed a web-based decision support tool for multiple hydrological hazards, such as floods and droughts, to discuss the decision-making process regarding budget, technicality, preparedness and response. For more information regarding the application of serious games in flood risk management, the reader is referred to Forrest et al. (2022)".

**Comment 2:** It could be helpful to give more meaningful abbreviations to E1, E2, E3 that indicate the type of information given, to help the reader more easily interpret the figures.

**Response**: Considering the suggestion of the Reviewer, we have given more meaningful abbreviations to E1, E2, E3 in the revised manuscript. Specifically, we now denote E1 as an experiment based on rainfall forecast + Quant_Pop, E2 based on rainfall forecast + Quali_Exp and E3 on rainfall forecast + Quali_Vul. We have made several additions in section 3.5 of

'Experimental Design', Table 2 and also added details of the experiments in Figure 2 of the revised manuscript to explain the modifications. Several texts in the Results and Discussion section have been modified according to the updated nomenclature. Below we present the added text in section 3.5 of 'Experimental Design' to make it clearer:

"Both rounds involve decisions based on three information sets each. Each information set contains hazard and vulnerability information. The hazard information is in the form of rainfall forecast, which is provided to the participants in either qualitative (Quali_RF) or quantitative (Quant_RF) form. The vulnerability information provided to the participants varies, which may be quantitative flood prone population (Quant_Pop), qualitative exposure (Quali_Exp) and qualitative vulnerability level (Quali_Vul). For easier understanding, we denote the use of each of the information sets as an experiment. As a result, Quant_RF + Quant_Pop is denoted as an experiment, E1, Quali_RF + Quali_Exp as E2 and Quali_RF + Quali_Vul as E3 in this study."

**Comment 3:** I suggest that you discuss the tradeoffs of what information can be made available (e.g. like relative time to prepare, ease or cost of availability, etc) in terms of why not suggest that participants are given all/best of the available information (i.e. high res rainfall, exposure, vulnerability).

**Response:** Thank you for the suggestion. We now discuss these tradeoffs in Section 3.3 ('Information provided to players') of the revised manuscript at lines 190-197:

"More detailed information, such as high-resolution rainfall and vulnerability, could potentially be available to decision-makers, and could thus have been provided to players. However, in India, having access to data, especially socio-economic data, is difficult because these are generally collected once in a decade, and are not available publically. For instance, the socio-economic and flood-related data used in this study have been obtained through the Municipal Corporation Greater Mumbai (MCGM) after a lengthy process. The raw socio-economic data obtained from the MCGM was processed to derive 15 vulnerability indicators. These were summarized by a color-code or value for an area because understanding and visualizing all 15 indicators to make a decision can be difficult, and also it may take too much time when quick decisions are required as in the case of floods. "

**Comment 4:** It would be helpful to expand on discussion about the 10% of participants that took actions in Gamma and were perceived to have misunderstandings. Could you learn anything from those participants that could improve the game?

**Response:** Thank you for the comment. Gamma was deliberately selected as a ward where the rainfall amount was much lower than in other wards and no flooding occurred (Figure 1b). The goal was to identify participants who could not understand the flood situation in the game or the rules (see lines 129-131 of the original manuscript). We were aware that some participants may not understand all the aspects of the game fully and hence play the game based on chance or by misunderstanding the rules of the game. The scores of such players would not reflect on the value of each information type. Therefore, we chose to remove those participants' responses from the analysis.

A deeper look into the decision-making sheets of these participants showed that the excluded participants made almost all the decisions for all three towns in the game. However, for some reason, they ended up making a few more decisions for Gamma than the other two towns. These participants could either be over-cautious in making the decisions, may not have understood the rules of the game or could not process the information provided to them for making decisions. This analysis is now added to the text in Section 4.1; lines 313 to 317 of the revised manuscript.

**Comment 5:** In discussing implications of results, the statement is made around line 315 that information on vulnerability helps make better decisions than just exposure. But yet just before this, it was stated that only in 1 round (R2) did vulnerability info yield the highest score. Please be clearer and not mis-leading/overly generalizing in your conclusions.

**Response**: We thank the Reviewer for pointing this out. Reconsidering this point, we do not see the relevance of the sentence, 'This suggests that the information of vulnerability helps to make better decisions compared to the information of exposure' in the present context at lines 314-315. We have removed the sentence from the revised manuscript.

**Detailed comments:**

**Comment 6:** pg2- line 25- in this paragraph, missing an obvious one of increasing population, thus more potential impacts.

**Response**:

Following the Reviewer's suggestion, we have added this aspect in the second paragraph of the Introduction section at lines 29-33 of the revised manuscript: "Provided that precipitation is the cause for a flood event, four main reasons may be advanced as to why the improvements in QPFs have not necessarily led to better mitigating the losses of lives and property. First, a growing population is associated with an increase in exposed lives and properties……"

**Comment 7:** pg5- mention somewhere in this section what type of players this is targeted to? e.g. educated audience that would play role of flood manager vs general public game scoring- it would be good to mention here that detailed information on decisions is in the appendix. Also, please clarify whether the managers on whom correct scores are based have 'hindsight' on what the best decision was based on what actually happened in the real event.

**Response:** The game is designed to be played by anyone who can understand the flood context and the game rules. Therefore, we targeted an educated audience who may understand the role of flood managers, the information they may have to process, and the decisions they may have to make. By being in academia, it was easier to access students enrolled in higher studies. Also, participants from research institutions ensure greater representation of social, cultural, and linguistic diversity in the game. We have added this in Section 3.6 (Game sessions and participants) of the revised manuscript.

Regarding *"detailed information on decisions is in the appendix"*. Following the Reviewer's suggestion, we now refer to Appendix B in Section 3.2.4 (Game scoring) to make it easy for the reader.

Regarding "*managers on whom correct scores are based have hindsight……*" – More specifically, the optimal decisions are based on the INSPIRE game rather than the flood event of 2005. The expert who provided the optimal decisions actually played the game to make the same decisions as the ones the participants had to make. The expert is a real decision-maker who works in the Municipal Corporation of Greater Mumbai (MCGM) which is the premier government department to make emergency decisions during a flash flood. The expert had no idea about the three towns in the game and their locations. We used the 2005 event only to provide a background to the INSPIRE city in the game, its geographical information, and socio-economic conditions, and to assess the hazard, exposure, and vulnerability information. We have added more

information regarding the background of the expert and the basis of the decisions in Section 3.2.4 (Game scoring) of the revised manuscript.

**Comment 8:** pg 5, line 144- it is mentioned that a modified precipitation forecast is used as the original severely underpredicted the actual event. This is worth more discussion later, given that flood managers cannot make good decisions when the information that they have ends up not at all aligning with the actualized event.

**Response:** We completely agree with the Reviewer, and that's why we chose to present the participants with a modified forecast. The forecast models were unable to predict the magnitude of rainfall which occurred during the 27-28 July 2005 event. There was a heavy rainfall forecast; however, it was far less than the actual rainfall that was received. In the game, we wanted to ensure that the participants made decisions based on an unbiased hazard forecast to test the different combinations of the hazard and vulnerability information and identify the best-suited combination for emergency decision-making.

The modified forecast used in the game is close to the observed rainfall. The observed rainfall was not used exactly to ensure that the rainfall in the three selected towns was contrasted, which allowed for playing out diverse decision-making contexts in the game. In reality, it is much more common to have biased forecasts. The effect of forecast biases on decision-making and forecast perception was explored in other studies (e.g. Arnal et al. 2016). We have added a discussion regarding the same in Section 3.3 (Information provided to players) of the revised manuscript at lines 172 to 182.

**Comment 9:** pg 6, line 158- this line noting '55% of its actual value' is confusing. Perhaps re-word to something like "each indicator value of Alpha was scaled by 55%.'

**Response:** Thank you for the suggestion. We have rephrased the following sentences in the revised manuscript at lines 204-205 as: "Each indicator is then *scaled* based on the fraction of the ward which was flooded during the 2005 event. For instance, close to 55% of Alpha's total area was flooded in the 2005 event, which implies that each indicator value of Alpha is *multiplied by 0.55* in this study."

**Comment 10:** pg7, line 185- it would be helpful to briefly explain how the indices (e.g. vulnerability) are calculated or their primary data inputs, rather than simply referring to a reference for all information.

**Response:** Taking into account the suggestion of the Reviewer, we have added the following paragraph in Section 3.4.4 (Exposure, sensitivity, adaptive capacity and vulnerability) revised manuscript at lines 233-235 – *"The normalized values of the indicators (Subsection 3.4.2) for each town are multiplied by their corresponding weights (Subsection 3.4.3) to calculate their weighted average. The values of the calculated sub-indices are then used to calculate the flood vulnerability index."*

**Comment 11:** pg9- line 251- it would be good to expand upon 'not overly straightforward' to connect to the fact that this demonstrates how flood managers can have trouble identifying the optimal outcome in the midst of the event.

**Response:** Thank you for the suggestion. We believe it will be a good idea to expand on this in the Discussion section (5.3) of the revised manuscript. Please see lines 484 to 490:

"Overall, the low scores of participants in the game indicate that emergency decision-making is not straightforward, especially in the case of sudden events such as flash floods. Considering that the game was designed in such a way that it closely represented the actual flood event of 2005, low scores demonstrate how flood managers can have trouble identifying the optimal outcome in the midst of an event. This also suggests how important it is that decision-makers are experts in their fields, have a comprehensive knowledge of their territory, and are well-trained to cope with difficult situations. Prior experience in decision-making can also help in analyzing the best possible options corresponding to the available resources and accuracy of information."

**Comment 12:** pg 10, line 274- when discussing how medians in R2 have shifted higher but tails are negative, should have 'but' instead of 'and' in 'located close together, and the tails of their distributions...'

**Response:** Thank you for pointing it out. In the revised manuscript, we have replaced 'and' with 'but' in the designated text.

**Comment 13:** Chart of E1,2,3 info could be helpful, like a more detailed version of Figure 2 left side.

**Response:** As suggested, we have revised Figure 2 to add more detailed information about E1, E2, and E3. Please see the revised Figure.

**Comment 14:** Figure 1- should be a bit larger so that sub-figure panels and associated text are more visible.

**Response:** We have enlarged the Figures of the manuscript. Please see the Figures of the revised version.

**References:**

Arnal L, Ramos M-H, Coughlan De Perez E, Cloke HL, Stephens E, Wetterhall F, Jan Van Andel S, Pappenberger F. 2016. Willingness-to-pay for a probabilistic flood forecast: a risk-based decision-making game. *Hydrol. Earth Syst. Sci*, 20: 3109–3128. https://doi.org/10.5194/hess-20-3109-2016.

Crochemore L, Ramos MH, Pappppenberger F, Van Andel SJ, Wood AW. 2016. An Experiment on Risk-Based Decision-Making in Water Management Using Monthly Probabilistic Forecasts. *Bulletin of the American Meteorological Society*. American Meteorological Society, 97(4): 541–551. https://doi.org/10.1175/BAMS-D-14-00270.1.

Forrest SA, Kubíková M, Macháč J. 2022. Serious gaming in flood risk management. *Wiley Interdisciplinary Reviews: Water*. John Wiley & Sons, Ltd, 9(4): e1589. https://doi.org/10.1002/WAT2.1589.

Sermet Y, Demir I, Muste M. 2020. A serious gaming framework for decision support on hydrological hazards. *Science of The Total Environment*. Elsevier, 728: 138895. https://doi.org/10.1016/j.scitotenv.2020.138895.

Terti G, Ruin I, Kalas M, Láng I, Cangròs I Alonso A, Sabbatini T, Lorini V. 2019. ANYCaRE: A role-playing game to investigate crisis decision-making and communication challenges in weather-related hazards. *Natural Hazards and Earth System Sciences*, 19(3): 507–533. https://doi.org/10.5194/nhess-19-507-2019.

**Comments from reviewer 2 and their response**

The paper by Singhal et al. describes a serious game mimicking the decision process during an extreme flood event. The game is based on the record floods that affected Mumbai in 2005. Overall, the paper is well written and relatively clear except for certain method points discussed below. The topic of using games to help understand and improve emergency management is highly relevant for the HESS journal in the global context of increased population in flood prone areas and changing climate. By establishing a certain distance between the players and reality, a game constitutes an efficient tool to extreme and often dramatic events. However, the game presented by the authors suffers from several fundamental flaws that make it unsuitable for publication in its present form. Two major flaws are discussed in the following section with more detailed comments provided in a subsequent part of the review report. All comments are numbered to facilitate later reference.

**>>> Major comments**

**Comment #1** - No considerations of ethic: serious games are qualified as "serious" because they are closely related to real situations and, hence, can have a powerful impact on their players. More generally, a serious game is a social experiment on human beings which requires a detailed assessment on the ethic of the process to ensure that players are protected from harm. The authors never mention this aspect which is surprising considering the policy of their respective institutions on this aspect (IISER, 2021; Université Grenoble Alpes, 2024). Following Fisher & Anushko (2008), ethical considerations (1) must address potential conflict of interest between the researchers and the participants, (2) must ensure informed consent of participants, (3) must ensure equitable treatment of participants regardless of their cultural or socio-economic background. We noticed several elements in the authors' game design that would require careful review in the light of these three principles:

We reply to the Reviewers' concerns in detail hereafter and propose to add a short paragraph in the methods and discussion sections to tackle this, repeating the information already present but in light of the point on ethics raised by the reviewer.

**(1.1)** The 2005 Mumbai floods was an extremely traumatic experience. There is a high risk of participants being negatively affected by the game if they were associated with the event. There is

no information in the paper on how the participants were identified, if they are voluntary, or if the purpose of the game was clearly explained to them.

**Response:** We appreciate the comment of the Reviewer. However, several of these aspects have been well covered in the original manuscript. We respond to each part of the comment sequentially.

Regarding the comment "high risk of participants being negatively affected by the game if they were associated with the event" – We refer the reviewer to our response to point 1.3.

Regarding the comment "There is no information in the paper on how the participants were identified, if they are voluntary" – We now mention in the manuscript that all game sessions were conducted in academic and research institutions (lines 297; 542). Extensive details about the institutions and the identified participants are also presented in Table B1. We also mention, in several instances in the manuscript, that the majority of participants are senior-level students, PhDs (48%), Masters (24%), and researchers (22%) at lines 235-236, 289-290 etc., and in Figure 3. Further, all the participants voluntarily participated in the game. We had taken due permission from the respective institutions to conduct the game session (please see the acknowledgment section in the manuscript), and participation in the game was entirely voluntary. To highlight the voluntary involvement of participants in the game, we have added a sentence in Section 3.6 (Game sessions and participants) at lines 303 of the revised manuscript

Regarding the comment "if the purpose of the game was clearly explained to them" – Utmost care was taken to ensure that the participants understood the context of the game, the scenarios, the information provided to them, the purpose of the game, the rules they had to play and the decisions they had to make (please see lines 106-107, 112-116, and 137-141 of the manuscript). In short, the participants were provided with all the information that is laid out in sections 3.2 and 3.3 of the manuscript. Also, we were cautious in ensuring that the game was made simple, yet most of the complications involved in decision-making were retained. To make it clearer, we have clarified this in Section 3.2.1 (Introduction of the game) at lines 125-127 of the revised manuscript.

**(1.2)** India is a country with a large cultural, linguistic and socio-economic diversity. It is not clear how this diversity was represented in the group of participants beyond their academic level. For example, the game seems to be based on questions asked in English with answers provided in the

same language through a spreadsheet. This favours disproportionally participants with an academic background such as PhD or researchers who, unsurprisingly, scored best in the game.

**Response:** We would like to highlight that we did not aim to cover the full socio-economic diversity present in India as we targeted participants with education levels and expertise that could lead them or have led them to become decision-makers in the context of floods. However, among this sub-group we considered including diverse social groups. One of the reasons why we chose national academic institutions to convene the game sessions was to ensure a diverse inclusion of social, cultural, and linguistic participation in the game. The three institutes where the game session was convened - *Central University of South Bihar*, *Indian Institute of Science Education and Research, Bhopal,* and *Indian Institute of Tropical Meteorology*, are well-reputed national institutes of India where students come from across the country for their higher education. These students belong to diverse cultural and socio-economic diversity; some may be poor or rich. To highlight this, we have added a text at lines 294-296 in Section 3.6 (Game sessions and participants) of the revised manuscript.

We were especially concerned about knowing the 'native city' of the participants as it is a better indicator of whether they have lived in a region prone to frequent floods. To this end, we explicitly asked the participants to share their 'native city and country' as can be referred to on page- 34, Section A of Figure A1 of the original manuscript. We share the regional representation of the participants in the Figure below.

Regarding the comment, "the game seems to be based on questions asked in English…….. scored best in the game", – The Reviewer has perhaps assumed that English is not the language in which students go about their education in India, which is not correct. English is the primary medium of education in India, especially in central state and national institutes. All exams, oral or written, research either theoretical or practical, are conducted in the English medium. The three institutes where we conducted the game function in the English language. The students of the three institutes write their exams in English; hence, we are confident that language was not a problem during the game. Hence, we assure the Reviewer that language had nothing to do with PhD or researchers scoring the best in the game. We have added a sentence, at lines 295-296, in Section 3.6 to emphasize that each session of the game is convened in the English language and the participants were duly informed about the same.

[Figure]

**Regional representation of participants**

(1.3) The participants are ignorant that the game is about Mumbai until this fact is revealed at the end of the game. This practice is a deceptive method which is highly debated in social sciences (Fisher & Fyrberg, 1994). Although not firmly prohibited, we are personally sceptical about its benefits due to the lack of trust it generates between participants and game organisers. This aspect may not be significant here due to the absence of on-going relationships between the authors and participants (the game seems to be a "one-off"). However, it could trigger problematic situations in relation to our point 1.1 above if participants suddenly realise that they have been playing with data from a flood and a city they are familiar with.

**Response:** The choice of not revealing that the game is based on an actual event in Mumbai is not deceptive in our case, as the Reviewer pointed out that the game is a "one-off". 'Deceptive' is perhaps a harsh word in this context. We firmly believe that revealing the actual backdrop of the game before the decision-making exercise would have actually created biases since many of the participants would have made the decisions based on their knowledge of the event rather than the information provided to them during the game. The game would have become more about how much the participants knew about the 2005 Mumbai event and less about what decision they made and why. We have clarified this in Section 3.1; lines 114-116 of the revised manuscript.

Based on their declaration (page 34, Section A of Figure A1 of the manuscript), 12 of the 123 participants who participated in the game belonged to Mumbai. We want to assure the Reviewer that during the debriefing session or even after, there were no problematic situations, none of the participants shared with us that the game negatively affected them by any means, or they developed a lack of trust in us. The one feedback we received from most of them was that the game presented a simplified nature of the actual events. This feedback suggests that, because participants were trained in the field, they had a tendency to see the event as a study case to learn from to improve methods developed in flood risk management rather than an emotional one. We have this feedback in cognizance and have acknowledged the same in the manuscript (please see lines 434-435 and 474-475 of the original manuscript).

**Comment #2** - Bias in research analysis: The method presented by the author suffers from several biases that could potentially affect their conclusions and limit their applicability to real-life emergency decision making. More specifically:

**Comment (2.1):** The authors excluded responses from participants that made more decisions for the "Gamma" district (Line 129 of the manuscript). This is not acceptable as it modifies the outcome of the game arbitrarily. There could be many reasons why participants decided to take such decisions. For example, they could have favoured economic interests in the "Gamma" district against population safety in other districts. Such decisions are morally questionable but they remain part of the game nonetheless.

**Response:** Excluding the responses from participants who made more decisions for Gamma was a deliberate effort to eliminate biases from overall results. Gamma was selected as one of the towns in the game with a clear motive to identify participants who could not understand the flood situation in the game or the rules to make the decisions (see lines 129-131 of the original manuscript). These participants would have played the game based on chance rather than their scientific conscience and would have eventually made some decisions correctly. The scores of these players would have negatively influenced the overall scores of all the players, which would not have been fair, nor would it have led to an accurate assessment of the decision-making abilities of other players. Additionally, the economic situation of Gamma was never given in the game and should not have entered the decision-making process.

Following the suggestion from the Reviewer, the answers from participants who took more decisions for Gamma were further analyzed. This analysis showed that the excluded participants made almost all the decisions for all three towns in the game. However, for some reason, they ended up making a few more decisions for Gamma than the other two towns. These participants could either be over-cautious in making the decisions, may not have understood the rules of the game or could not process the information provided to them for making decisions. This analysis is now added to the text in Section 4.1; lines 313 to 317 of the revised manuscript.

**Comment (2.2):** The game duration is extremely short with 25 minutes for the two rounds and an additional 15 minutes of questions and discussion. In addition, the game is played individually without any interaction between the participants except during the last 15 minutes. Consequently, the game does not explore human interactions and coordination at all, which are fundamental in analysing emergency response (Drabek, 1985).

**Response:** The participants were given 25 minutes to make the decisions. The time spent on providing the background information regarding the study area, the structure of the two rounds, and the rules to play the game are not included in these 25 minutes. We wanted to keep the game time short. The game was structured in such a way that it should not take more than 60 minutes to complete one game session (including game background, rules, game-play, and debriefing). We also pilot-tested the game with several volunteers and the actual decision-making did not go beyond 25 minutes. Also, in none of the sessions, participants demanded or wished for any extra time. Generally, situations of flash floods in a metropolitan area require quick response and action. Sharifzadeh et al. (2020) reviewed 101 serious games in the health sector and reported the most common gameplay duration was 30-45 min. We have clarified the duration of the game in Section 3.2.2 (Game rounds) at lines 135-137 of the revised manuscript.

Regarding the comment – "The game is played individually without any interaction between the participants" – In the original manuscript, we have mentioned the non-interactive nature of the game as one of the limitations (see lines 425-426). However, we explain the rationale here. In our study, we did not intend to explore the interaction of participants during decision-making. There are several serious games in the literature which have already accounted for human interactions in the game (Rusca *et al.*, 2012; Terti *et al.*, 2019; Bakhanova *et al.*, 2020; Neset *et al.*, 2020). We did not aim to understand how humans interact in their roles to make decisions.

Rather, our aim is to study how hazard-vulnerability information that would be communicated by a meteorological department or an early warning system is received and interpreted by an individual to appraise the criticality/priority of a situation. We have added a few lines to clarify this in Section 5.5; lines 512-515 of the revised manuscript. The collaborative work should come in a later step. We could have included the component of role-playing to make the decision-making interactive, however, we felt it was not necessary considering the objectives of the study.

**Comment** (2.3) All decision variables are colour coded, which removes the ability for the participants to weight quantitatively the information provided. We appreciate the author's intent to simplify the information and allow the participants to compare disparate data. However, this is not the reality of an emergency decision process where flood managers must deal with sometimes confusing data.

**Response:** We would like to point out that not all decision variables in the game are color-coded. The 'rainfall forecast' and 'flood-prone population density' provided to the participants are quantitative (please refer to lines 139-140, 218, 223, Table 2 and Figure 2 of the original manuscript). In addition, it is precisely the aim of this paper to explore whether color-coded information may be acceptable to base decisions on as opposed to quantitative data (sometimes it could be a large amount of data). This assumption is one that has been explored in other scientific fields such as the health sector (Goldman's algorithm; Qamar et al. 1999). This study showed that a limited amount of information tailored for a specific decision (here, a decision tree) was sometimes more desirable than a large amount of quantitative information, especially in contexts that require quick decisions, as is the case during floods or in this game. We were curious to explore this in the context of disaster management.

Regarding the color-coded form of information, the Reviewer commented that it 'removes the ability for the participants to weight quantitatively the information provided'. We would instead argue that it's the quantitative form of information that generally makes it difficult to weigh the information and not the qualitative one. Here, qualitative information is considered because one can easily distinguish between the variables based on the colors. The colors (say green, yellow, and red) are classified based on a particular scale (say low, medium, and high), which makes it easier to interpret. On the other hand, to weigh the quantitative form of information one has to understand what the numbers mean. For instance, a decision-maker who does not have a strong

understanding of precipitation amounts will find it difficult to understand what 10 mm, 25 mm, 50 mm, and 100 mm mean in terms of their severity and potential impacts. However, precipitation amounts assigned colors such as green, blue, orange, and red make it easier for the decision-maker to make better decisions.

Regarding the comment that 'flood managers must deal with sometimes confusing data' – We agree with the Reviewer and that is why we are trying to investigate through this game whether complex information about hazard, exposure and vulnerability can be simplified to make emergency decisions. Results show that participants made better decisions with the qualitative form of information compared to the quantitative form.

**Comment** (2.4) Vulnerability data are presented to participants at the same time or even after rainfall forecast data. The game setup seems to reproduce the case of an untrained manager going through her or his very first flood and who discovers vulnerability hot spots at the same time than rainfall forecasts arrive. This is not realistic for a seasoned manager who knows the city well. We suggest reconsidering this point and present the vulnerability data well in advance to the players so that they understand the layout of the city before the game starts. The lack of context understanding seems to be confirmed by the game results where participants obtained better score in the second round compared to the first (see Line 254).

**Response:** We would like to clarify several points in order to reply to this comment. We would like to clarify each misunderstanding sequentially.

Regarding the sub-comment "Vulnerability data are presented to participants at the same time or even after rainfall forecast data" – We would like to clarify to the Reviewer that the vulnerability data is presented to the participants at the same time as the rainfall forecast information, and not later.

Regarding the sub-comment "The game setup seems to reproduce the case of an untrained manager going through her or his very first flood and who discovers vulnerability hot spots at the same time than rainfall forecasts arrive. This is not realistic for a seasoned manager who knows the city well" – In the game, the town is first presented simply, which serves as an introduction to the territory at stake. It is then explained that an imaginary crisis unit provides participants with different kinds of information based on which the flood risk manager (participant) makes the decisions. It should be common for any flood emergency meeting to bring all the available

information to the discussion, including vulnerability. It would be unreasonable for decision-makers to remember all the vulnerability information about the whole city, particularly one of the size of Mumbai, and expect them to make decisions just based on rainfall forecast. Furthermore, we have not mentioned in the manuscript that the decision-maker 'discovered' vulnerability for the first time. It is not realistic for a manager to consider all the available information while making decisions, especially in a city in which close to 20 million people live. Moreover, there may be a strong case that the vulnerability of the city is constantly updated. The manager must have the latest information, including that of vulnerability, on his or her table while making decisions. According to us, a flood manager who makes decisions based on its historical understanding of the city's vulnerability is not untrained, but rather careless.

Regarding the comment "We suggest reconsidering this point and present the vulnerability data well in advance to the players so that they understand the layout of the city before the game starts"-We do inform the participants about the area they will have to manage, with relevant geographical and socio-economic characteristics prior to the game and the decision-making (see lines 106-107 of the original manuscript). We also provide them with certain fields of information, such as the water level of the lake and river, sea tide height, prevailing ground situation, and possible future developments before the rounds of the game start (see lines 113-114 of the original manuscript). We believe this information helped the participants to understand the layout of the city before the game started.

**Comment** (2.5) The game assumes that there is a "correct" answer for every round defined by local experts. This aspect is quite disturbing as it is difficult to know what the best decision in a city as complex as Mumbai is when facing a flood as extreme as 2005. In addition, there is little information about who the experts are and if participants accept them as experts whereas the definition of a correct decision in this case is likely to be highly contested. We suggest considering more diverse form of rewards such as achieving consensus (if debate was allowed between participants) or showing consistency throughout the game (an important quality of emergency decisions).

**Response:** We would like to clarify that the optimal decisions are not based on the 2005 extreme event. They are based on the game that we have designed in this manuscript. The expert who provided the optimal decisions actually played the game to make the same decisions as the ones

the participants have to make. The expert is a real decision-maker who works in the Municipal Corporation of Greater Mumbai which is the premier government department to make emergency decisions during a flash flood. The expert had no idea about the three towns in the game and their locations. We used the 2005 event only to provide a background of the INSPIRE city in the game, its geographical information, socio-economic conditions and to assess the hazard, exposure and vulnerability information. We have added more information regarding the background of the expert and the basis of the decisions in Section 3.2.4 (Game scoring) at lines 151-160 of the revised manuscript.

Regarding the comment "if participants accept them as experts whereas the definition of a correct decision in this case is likely to be highly contested" – We have obtained optimal decisions for the game from a real decision-maker in Mumbai who goes through the decision-making process regularly. We don't believe it would be right to understand from the participants whether they accept the decisions or not. In reality, the general public does not generally contest the decisions of the decision-makers. However, we do recognize that several decision sets may be equally correct. This decision set by the expert decision-maker merely serves as a reference to compare participants' answers. To make it clearer, we have added more information regarding it in Section 3.2.4 (Game Scoring) of the revised manuscript.

>>> **Minor comments**

**Comment 3:** Line 25, "Three main reasons may be …":  a fourth more fundamental reason is simply that  extreme rainfall do not necessarily translates into high hazard. There are hydrogical (e.g. antecedent conditions, non-linear runoff generation, ...) and hydrodynamic (e.g. topography, levee systems, backwater effects, ...) factors that complicate flooding processes and reduce the value of rainfall information.

**Response:** We thank the Reviewer for the suggestion. We have modified this list of reasons to include suggestions from both reviewers in the Introduction Section at lines 27-32 of the revised manuscript: "Despite the increasing availability and performance of QPFs across the globe, loss of lives and economic damage has continued to rise (Nanditha and Mishra, 2021; Lala et al., 2021; Singhal et al., 2022). The first reason is simply that extreme precipitation does not necessarily lead to a flood hazard, which can, for instance, be explained by hydrological (e.g., rather dry antecedent conditions) or hydrodynamic (e.g., structural mitigation measures) factors. Provided that

precipitation is the cause of a flood event, four main reasons may be advanced as to why the improvements in QPFs have not necessarily led to better mitigating the losses of lives and property. First, a growing population is associated with an increase in exposed lives and properties. Second, despite improvements, the QPFs still lack the quality to accurately predict the magnitude, intensity, and duration of extreme hazards (EPE or flash floods). Third, early warning systems have generally used QPFs to focus on hazards rather than on their impacts at the local scale. Lastly, the obtained hazard information is not well integrated with the information of local exposure and vulnerability."

**Comment 4:** Line 85, "Gupta and Nair": the reference is not about the Mumbai flood but about floods in Chennai and Bangalore. Please remove this reference and replace it by a more appropriate one.

**Response:** Thank you for the suggestion. We have removed the reference "Gupta and Nair" from the revised manuscript and replaced it with (Gupta, 2007).

**Comment 5:** Line 105, "Flood Manager": this role needs to be defined in greater details. There is a great diversity of flood managers ranging from liaison officers to operators of major infrastructures. Please clarify this point and explain how it was presented to the participants.

**Response:** The main role of the "Flood Risk Manager" was to make the best possible emergency decisions to minimize the impact of the extreme precipitation and flood. The manager led a fictitious team of representatives from the Meteorological Department, Department of Town Planning, Department of River Management, Department of Coast Management and the media cell (please see lines 114-115).

**Comment 6:** Line 115, "Meteorological Department, Department of Town Planning, Department of River Management, Department of Coast Management and the media cell": why are there so many organisations providing information and only one role for the participants (Flood Manager)? Please clarify why it is important to distinguish the information provider and its effect on the responses during the game.

**Response:** These organizations have been included in the game to make it more interesting for the participants and to keep the game closer to reality. In the event of flash floods, generally, a meeting is called where representatives from different departments discuss a variety of information before

making any decision. In our game, the Flood Manager is the head of the crisis management unit. The names of different departments which provide the information have no effect on the responses of the participants. Even if the names were not included in the manuscript, the participants would have received this information. However, the names of these departments make it easier for the participants to understand and remember the information provided.

**Comment 7:** Line 142, "The accumulated rainfall forecast, used in the game, is a slight modification": Please clarify if there was an attempt to reproduce the skill level of recent rainfall forecast. This information is important to assess if the forecasts are realistic for current decision making in Mumbai.

**Response:** We would like to clarify that there was no attempt to reproduce the skill level of recent rainfall forecasts. Instead, we used the observed rainfall for the particular event as an unbiased forecast to which we add uncertainty. The actual rainfall forecasts were highly underestimating the observed rainfall (lines 144-145). The best forecast (UKMO) predicted 120-160 mm (lead time day 3), 280-320 (lead time day 2), and 200-240 (lead time day 1), as reported by Bohra et al. (2006). The difference between the forecast and observed rainfall was such that it would have been incredibly difficult for any post-processing technique to match the observation. Here we precisely decided to focus on forecasts that are close to unbiased in order to focus on the influence of the vulnerability information. To clarify this, we have added more information in Section 3.3 of the revised manuscript at lines 172-182.

**Comment 8:** Line 147, "The information of exposure and vulnerability is statistically calculated": this sentence is not clear. Remove this statement and refer to the following sections which explain the process.

**Response**: As suggested by the Reviewer, we have removed the statement from the revised manuscript and instead refer to section 3.4.

**Comment 9:** Line 152, "Vulnerability and Exposure analysis": the concept of exposure is confusing. As indicated by the authors and following Gallopín (2006), vulnerability can be decomposed into exposure, sensitivity and adaptive capacity. Consequently, exposure is a part of vulnerability, not an independent concept. However, the section title at line 152 suggests that it is distinct. Please clarify.

**Response:** The reviewer is correct that vulnerability can be divided into exposure, sensitivity and adaptive capacity. To avoid any misunderstandings, we have replaced the existing Section title 3.4 from 'Vulnerability and Exposure analysis' to just 'Vulnerability analysis'.

**Comment 10:** Line 156, "standardized": please remove this word. The authors are simply calculating the value of each indicator based on the proportion of area flooded in the ward assuming an homogeneous distribution of the indicator across the ward. Standardized has a different meaning which often relates to subtracting the mean and dividing by the standard deviation.

**Response:** In the revised manuscript, we have used the word 'scaled' instead of 'standardized'. The following sentences have been rephrased in the revised manuscript: "Each indicator is then *scaled* based on the actual area of the ward which was under flood during the 2005 event. For instance, close to 55% of Alpha's total area was flooded in the 2005 event, which implies that each indicator value of Alpha is *scaled by* 55% of its actual value in this study." Please see Section 3.4; line 202 of the revised manuscript.

**Comment 11:** Line 160, "normalized": Please define this normalisation.

**Response:** We have mentioned the use of maxima-minima method for normalization in Section 3.4 of the revised manuscript. We have also defined the method in Section 3.4.2 at lines 216-208 of the revised manuscript as:*"The maxima – minima method scales the value of an indicator between 0 and 1. The minimum value of the indicator is subtracted from the value of a selected indicator which is then divided by the range of the indicator"*.

**Comment 12:**Line 193, "based on the beta distribution": this approach seems overcomplicated for the definition of simple indicators. The use of the beta distribution adds the uncertainty associated with the choice of the distribution and its parameter values. We suggest replacing this by the quantiles of the indicators across the 24 wards.

**Response:** We disagree with the Reviewer in this context. Statistically, a quantile (or any statistics) starts being robust for a sample size of 30 or greater, hence the choice of the beta distribution in our study. The method is not over complicated and has been used in several vulnerability studies (Carrão *et al.*, 2016; Byers *et al.*, 2018; Singhal and Jha, 2021; Tanim *et al.*, 2022).

**Comment 13:** Line 220, "qualitative rainfall forecasts": please clarify how are rainfall forecasts color coded.

**Response:** Since the extreme rainfall event witnessed over Mumbai in a few hours was unprecedented, there are no existing criteria that can be used to classify that amount of rainfall. Hence, we developed criteria for classifying the rainfall forecast as color codes in the manuscript. First, historically observed rainfall amounts were explored to find the highest ever 1- 1-hourly and 3- 3-hourly rainfall over Mumbai city. These rainfall amounts were 113 and 253 mm, respectively. The rainfall amounts were then classified into four categories based on equal proportions.

If Rainfall (mm) <=113, the category is defined as Level I (green),

If Rainfall (mm) >113 to 183, the category is Level II (yellow)

If Rainfall (mm) >183 to 253, the category is Level III (orange)

If Rainfall (mm) >253, the category is Level IV (red)

We have added the criteria as Appendix C; page 39 in the revised manuscript. The same has also been highlighted in Section 3.3 (Information provided to players) of the revised manuscript.

**Comment 14:** Line 302, "level of education does play a role in decision-making": it is not obvious that researchers are best placed to take high risk decisions under intense time pressure. We believe that this statement is in fact the result of the multiple biases introduced by the game described in the previous section.

**Response:** We beg to disagree with the Reviewer. As mentioned, the bias in the population of participants was deliberate. Among these participants, those with experience performed better. We hope that our responses have clarified the reasons for the chosen biases and explained the other points raised. We have modified the title of Section 4.3 to "the role of education and experience in emergency decision-making" and rewritten the phrase as "level of experience does play a role in decision-making" at several instances in Section 4.3 of the revised manuscript.

**>>> References**

Drabek, T. E. (1985). Managing the Emergency Response. Public Administration Review, 45, 85–92. https://doi.org/10.2307/3135002

Fisher, C. B., & Anushko, A. E. (2008). Research ethics in social science. The SAGE Handbook of Social Research Methods, 95–109.

Fisher, C. B., & Fyrberg, D. (1994). Participant partners: College students weigh the costs and benefits of deceptive research. American Psychologist, 49(5), 417–427. https://doi.org/10.1037/0003-066X.49.5.417

Gallopín, G. C. (2006). Linkages between vulnerability, resilience, and adaptive capacity. Global Environmental Change, 16(3), 293–303. https://doi.org/10.1016/j.gloenvcha.2006.02.004

IISER. (2021). Manual on R&D Project Management with Guidelines (p. 82). Indian Institute of Science Education and Research Bhopal. https://www.iiserb.ac.in/assets/all_upload/pdf/548351c53d6600ee2db8ebb02b804208.pdf

Université Grenoble Alpes. (2024). Le comité d'éthique et de déontologie. https://www.univ-grenoble-alpes.fr/universite/engagements/ethique-et-deontologie/le-comite-d-ethique-et-de-deontologie-1145514.kjsp?RH=1665562627143

**References**

Bakhanova E, Garcia JA, Raffe WL, Voinov A. 2020. Targeting social learning and engagement: What serious games and gamification can offer to participatory modeling. *Environmental Modelling & Software*. Elsevier, 134: 104846. https://doi.org/10.1016/J.ENVSOFT.2020.104846.

Byers E, Gidden M, Leclere D, Balkovic J, Burek P, Ebi K, Greve P, Grey D, Havlik P, Hillers A, Johnson N, Kahil T, Krey V, Langan S, Nakicenovic N, Novak R, Obersteiner M, Pachauri S, Palazzo A, Parkinson S, Rao ND, Rogelj J, Satoh Y, Wada Y, Willaarts B, Riahi K. 2018. Global exposure and vulnerability to multi-sector development and climate change hotspots. *Environmental Research Letters*. IOP Publishing, 13(5): 055012. https://doi.org/10.1088/1748-9326/AABF45.

Carrão H, Naumann G, Barbosa P. 2016. Mapping global patterns of drought risk: An empirical framework based on sub-national estimates of hazard, exposure and vulnerability. *Global Environmental Change*. Pergamon, 39: 108–124. https://doi.org/10.1016/J.GLOENVCHA.2016.04.012.

Gupta K. 2007. Urban flood resilience planning and management and lessons for the future: a case study of Mumbai, India. *Urban Water Journal*. Taylor & Francis, 4(3): 183–194. https://doi.org/10.1080/15730620701464141.

Neset TS, Andersson L, Uhrqvist O, Navarra C. 2020. Serious gaming for climate adaptation—assessing the potential and challenges of a digital serious game for urban climate adaptation. *Sustainability (Switzerland)*, 12(5): 1–18. https://doi.org/10.3390/su12051789.

Qamar A, McPherson C, Babb J, Bernstein L, Werdmann M, Yasick D, Zarich S. The Goldman algorithm revisited: prospective evaluation of a computer-derived algorithm versus unaided physician judgment in suspected acute myocardial infarction. Am Heart J. 1999 Oct;138(4 Pt 1):705-9. doi: 10.1016/s0002-8703(99)70186-9. PMID: 10502217.

Rusca M, Heun J, Schwartz K. 2012. Water management simulation games and the construction of knowledge. *Hydrology and Earth System Sciences*, 16(8): 2749–2757. https://doi.org/10.5194/HESS-16-2749-2012.

Sharifzadeh N, Kharrazi H, Nazari E, Tabesh H, Edalati Khodabandeh M, Heidari S, Tara M. 2020. Health Education Serious Games Targeting Health Care Providers, Patients, and Public Health Users: Scoping Review. *JMIR serious games*. JMIR Serious Games, 8(1): e13459.

https://doi.org/10.2196/13459.

Singhal A, Jha SK. 2021. Can the approach of vulnerability assessment facilitate identification of suitable adaptation models for risk reduction? *International Journal of Disaster Risk Reduction*. Elsevier Ltd, 63(July): 102469. https://doi.org/10.1016/j.ijdrr.2021.102469.

Tanim AH, Goharian E, Moradkhani H. 2022. Integrated socio-environmental vulnerability assessment of coastal hazards using data-driven and multi-criteria analysis approaches. *Scientific Reports 2022 12:1*. Nature Publishing Group, 12(1): 1–28. https://doi.org/10.1038/s41598-022-15237-z.

Terti G, Ruin I, Kalas M, Láng I, Cangròs I Alonso A, Sabbatini T, Lorini V. 2019. ANYCaRE: A role-playing game to investigate crisis decision-making and communication challenges in weather-related hazards. *Natural Hazards and Earth System Sciences*. Copernicus GmbH, 19(3): 507–533. https://doi.org/10.5194/NHESS-19-507-2019.